# Commercial influenza vaccines vary in HA-complex structure and in induction of cross-reactive HA antibodies

Mallory L. Myers[1,5], John R. Gallagher [1,5], Alexander J. Kim[1], Walker H. Payne[1], Samantha Maldonado-Puga[1], Haralabos Assimakopoulos[1], Kevin W. Bock [2], Udana Torian[1,3], Ian N. Moore[2,4] & Audray K. Harris [1] ✉

Influenza virus infects millions of people annually and can cause global pandemics. Hemagglutinin (HA) is the primary component of commercial influenza vaccines (CIV), and antibody titer to HA is a primary correlate of protection. Continual antigenic variation of HA requires that CIVs are reformulated yearly. Structural organization of HA complexes have not previously been correlated with induction of broadly reactive antibodies, yet CIV formulations vary in how HA is organized. Using electron microscopy to study four current CIVs, we find structures including: individual HAs, starfish structures with up to 12 HA molecules, and novel spiked-nanodisc structures that display over 50 HA molecules along the complex's perimeter. CIV containing these spiked nanodiscs elicit the highest levels of heterosubtypic cross-reactive antibodies in female mice. Here, we report that HA structural organization can be an important CIV parameter and can be associated with the induction of cross-reactive antibodies to conserved HA epitopes.

Influenza is a widely prevalent respiratory virus with pandemic potential. Healthy individuals, even those who are vaccinated, are often still susceptible to influenza infection and may become sick, disrupting work and well-being[1]. At-risk individuals that may be immunologically suppressed due to age or specific health issues can develop life threatening infections[2]. Yet even prime aged individuals may fall victim to seasonal influenza, especially with the lack of supportive care for these infections[3]. When new zoonotic strains are created by reassortment between animal and human lineages, antigenically novel influenza viruses have caused pandemics which have put even the healthy populations at great risk[4]. The commercial vaccine market for seasonal influenza vaccines has become a billion dollar industry[5], in part due to the rapid variation of influenza antigens.

Influenza is an enveloped virus, presenting three antigens on the viral surface. Hemagglutinin (HA), the most prevalent viral glycoprotein, is responsible for binding to target cell glycans, as well as execution of membrane fusion. Neuraminidase (NA) cleaves cellular glycans to assist in viral release from host cells, counteracting the binding activity of HA. Matrix (M2) is an ion channel embedded in the viral envelope which is integral for pH homeostasis of the virus[6–8]. Following either natural infection with influenza virus or immunization with HA-based vaccines, the majority of protective antibodies are targeted to HA[9–11]. HA is expressed as a homotrimer consisting of three copies of the protein HA0. Each HA0 protomer is later proteolytically cleaved into two fragments: HA1 and HA2. HA1 is the N-terminal fragment and comprises the head domain, which is the membrane-distal region of the 13 nm tall viral glycoprotein. HA1's protein sequence also contributes to the stem domain at the opposite end of the HA protein from the head and nearer to the viral envelope. HA2 is the C-terminal fragment of HA0 cleavage, and all residues of HA2 contribute to the

[1]Structural Informatics Unit, Laboratory of Infectious Diseases, National Institute of Allergy and Infectious Diseases, National Institutes of Health, 50 South Drive, Room 6351, Bethesda, MD 20892, USA. [2]Infectious Disease Pathogenesis Section, National Institute of Allergy and Infectious Diseases, National Institutes of Health, 33 North Drive, Room BN25, Bethesda, MD 20892, USA. [3]Present address: Laboratory of Human Carcinogenesis, National Cancer Institute, 37 Convent Drive, Room 306C, Bethesda, MD 20892, USA. [4]Present address: Yerkes National Primate Research Center, Emory University, 954 Gatewood Rd NE, Atlanta, GA 30329 37, USA. [5]These authors contributed equally: Mallory L. Myers, John R. Gallagher. ✉e-mail: harrisau@mail.nih.gov

stem domain. HA2 contributes the most conserved amino acids in HA[12], which perform the act of fusing host and viral membranes during viral entry. Because the HA head is both immunodominant and highly variable, yearly reformulation of influenza vaccines is required to ensure that HA antigenic variation does not escape vaccine-elicited antibodies[13,14].

Historically, influenza vaccine development was evaluated by antigenicity and protection in animal models focusing on the concentration of HA, but recently other viral components such as NA and M2 are being explored[15–17]. Commercially available influenza vaccines have been widely accessible in the United States since the 1950s[18] with the CDC regulating the concentration of HA in those offered[9]. Early vaccine formulation strategies and procedures, many of which are still in use, required generation of antigen via seed viruses that are then used to infect embryonated chicken eggs, harvest virus, and extract HA by methods of varying specificity[19]. To avoid the problem of egg-adapted HA mutations, new strategies have been developed to manufacture vaccines directly from mammalian cells infected with influenza[20]. Further developments in vaccine manufacturing were seen with the approval of an MF59 adjuvanted commercial vaccine[21], while another vaccine formulation utilized recombinant HA derived from a baculovirus expression system in insect cells[22–24]. To date, structural characterization of commercial HA subunit vaccines for influenza has not extended beyond negative-stain electron microscopy (EM)[25]. Those studies showed HA presented in rosette formations when isolated from inactivated virus or produced by recombinant baculovirus expression[23,25,26]. Rosettes were naturally formed by removing lipid components, resulting in the transmembrane domain of multiple HA proteins being constricted into small lipid micelles. Within rosettes, the accessibility of the HA head domain was unperturbed, causing no concern for decreased immunogenicity as vaccine development had focused on the antigenically dominant head domain of HA. However, recent universal influenza vaccine initiatives have shifted attention from a singular focus on HA head-targeted antibodies toward consideration of HA stem epitopes.

Pursuit of novel HA stem vaccines aiming to protect against both current and future influenza viral strains ignited interest in targeting the conserved stem of HA, which is composed of evolutionarily conserved sequence of HA2 and contains residues essential for viral fusion[27,28]. Presently, differences in the extent that existing influenza vaccine formulations display influenza virus stem epitopes has not been fully characterized. While some formulations offer better cross-protection against influenza viral strains that are not present in the vaccine, the mechanism for this effect in terms of HA organization and conserved epitope display is unknown[21,29]. The addition of an adjuvant to a vaccine is known to directly stimulate the immune response in additional ways to subunit vaccines[30]; however, adjuvant can also profoundly affect the higher-order organization of antigen in solution.

In this work, we explore the organization of HA within four licensed influenza vaccines: Fluzone High Dose (HD), Flucelvax, Flublok, and Fluad, by EM to identify structural correlates to the induction of antibodies against different influenza virus HA subtypes. We show that the four commercial influenza vaccines have HA components organized in heterogenous polymorphic complexes when examined by 2D imaging and classification by negative stain EM. While different in organization, all vaccines display the conserved stem HA epitopes and bind to broadly reactive human anti-stem antibodies FI6v3 and CR6261. However, the four vaccines have important differences in their ability to elicit cross-reactive antibodies to HA subtypes not part of their formulation. All of the vaccines protect against a homologous H1N1 challenge, while only one vaccine, Fluad, elicits significant levels of antibodies to the stem and for HA subtypes not present in the vaccine formulation. This vaccine also presents a novel spiked-nanodisc HA structure. Analysis of the spiked nanodiscs indicates that HA stem epitopes are more readily accessible, suggesting a

structural basis for the ability to induce heterospecific antibodies to subtypes not present in the vaccine. Induction of heterospecific antibodies will be important for the development of both more efficacious seasonal and universal influenza vaccines that elicit broad immune responses to antigenically divergent HAs.

## Results

### HA organization in different commercial vaccines

Negative-stain EM was used to determine if antigen structure and organization within the four commercial influenza vaccines were different. Of the four commercial vaccines studied, Fluad and Fluzone High Dose (HD) are trivalent, containing HA antigens for H1 and H3 subtypes for influenza A virus, and B/Victoria lineage for influenza B virus. Flublok and Flucelvax are quadrivalent vaccines due to their composition including an additional influenza B virus B/Yamagata strain. Specific influenza strain composition for each vaccine is given in Supplementary Fig. 1a. All vaccine HA antigens were generated from inactivated influenza virus except for the Flublok vaccine, which was composed of recombinant HA protein (Supplementary Fig. 1b). Western blot analyses of the vaccines' components confirmed the presence of the major antigens of the commercial vaccines (Supplementary Fig. 2a–c), including HA proteins of H1 and H3 subtypes in addition to influenza B virus HA glycoproteins. Other viral components were present at lower levels and exclusively in virus-derived formulations. These other viral components included neuraminidase (NA), matrix, and nucleoprotein (Supplementary Fig. 2d–f).

Four configurations summarized the organization of HA in these vaccine formulations: (1) isolated HA molecules, (2) starfish-like complexes with HA radiating from a central core, (3) HA in micelles, and (4) HA in a larger disc-like structure (Fig. 1, Supplementary Fig. 3). In EM micrographs, glycoproteins appear as a lighter shade of gray against a darker background, since the negative stain is excluded by presence of the protein. On rare occurrences, genomic ribonucleoprotein (RNP) complexes were also visible (Fig. 1a, asterisk). Class-averages were generated from thousands of picked particles in order to visualize the most common structures observed across the entire sample. In general, average structures indicated that the conformation of HA was presented in the characteristic peanut-shapes of prefusion trimeric molecules, which were in multimeric starfish-like complexes (e.g. Fig. 1b, left, middle panels) and as isolated HA trimers associated with small volumes of membrane (e.g. Fig. 1b, right panel).

Starfish-like structures were characterized by glycoprotein spikes protruding from a central location containing HA transmembrane domains. Starfish-like complexes of HA (Fig. 1a, black arrows) and isolated HA trimers (Fig. 1b, right panel) were present in Fluzone HD. Flucelvax also contained both isolated HAs and starfish complexes, but differed from Fluzone HD in that isolated HA molecules were more prevalent (Fig. 1c, white arrows) than starfish-like complexes (Fig. 1c, black arrows). Class averages of Flucelvax indicated the presence of HA-starfish-like complexes, isolated HA molecules, and NA heads which were identifiable by their tetrameric symmetry (Fig. 1d, right panel). Unique to Flucelvax were relatively large 100-200 nm lipid structures (i.e. micelles) with sparsely distributed HA proteins protruding from their surface (Fig. 1c, black bracket). Flublok was the most structurally homogeneous, with HA organized exclusively in starfish-like complexes, commonly containing 5-12 HA trimers (Fig. 1e, f). To address if individual HA molecules were in prefusion or post-fusion conformation, individual HA trimers were manually picked to select proteins that were sufficiently isolated from other entities in the EM images. These individual HA trimers were averaged to improve resolution, and then measured for comparison to previously reported structures. We found that all influenza vaccines yielded HA profiles that were consistent with prefusion HA conformation, based upon the height of the HA trimer, and based upon the shape profile (Supplementary Fig. 4).

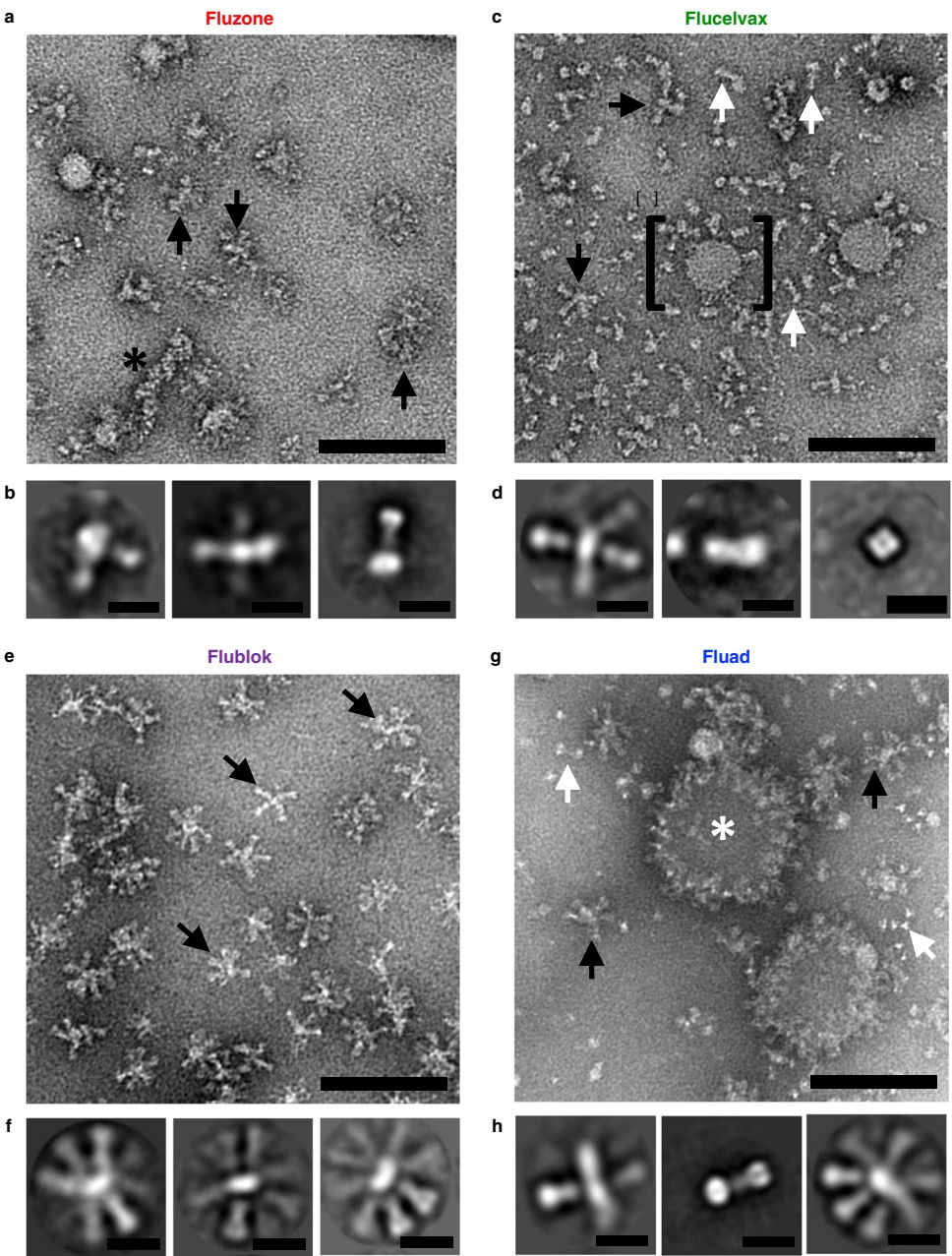

**Fig. 1 | Structural organization of HA antigens in influenza vaccines. a** Images from negative-stain EM of Fluzone HD vaccine and **b** select 2D classes averages. Similarly, images and class averages for **c**, **d** Flucelvax, **e**, **f** Flublok, and **g**, **h** Fluad. Complexes observed include HA starfish-like complexes (black arrows), isolated molecules (white arrows), large round complexes (white asterisk), RNP (black asterisk) and small round mostly bald particles (bracket, putative micelles). Scale bars are 100 nm for panels **a**, **c**, **e**, and **g**, and 10 nm for panels **b**, **d**, **f**, **h**. Grids representing each vaccine were prepared at least three times, to account for staining variation.

While Fluad contained isolated HA molecules and HA-starfish complexes similar to other vaccines (Fig. 1g, h), Fluad also contained an additional component not seen in other formulations. Large round complexes were observed with glycoprotein spikes dotting their outer perimeter (Fig. 1g, white asterisk). Many HA proteins on these complexes were visible as top-views that visualized the HA three-fold symmetry axis as a triangular arrangement of HA protomers. Different years of each of these influenza subunit vaccines were compared and the structures for each vaccine matched the EM data of Fig. 1. Thus, confirmatory EM results indicated that the organization of HA molecules was consistent from year to year (data not shown).

## Modeling the effects of HA stem epitope crowding

The observed differences in HA molecule spacing and arrangement across different vaccine complexes led us to investigate the likelihood that HA stem epitopes were blocked or occluded by steric hindrance from neighboring HA molecules. We enumerated possible pairwise HA orientations by Monte Carlo simulation and evaluated if these configurations would occlude stem antibody binding (Fig. 2a, b). We modeled the interaction with fragment antigen binding (Fab) domain instead of full IgG, thus our findings represent an upper bound to the number of sterically permitted binding conformations. IgG molecules have flexibility between domains[31], which allows them to deviate from a rigid Y-shaped conformation. This flexibility enables the approximation of

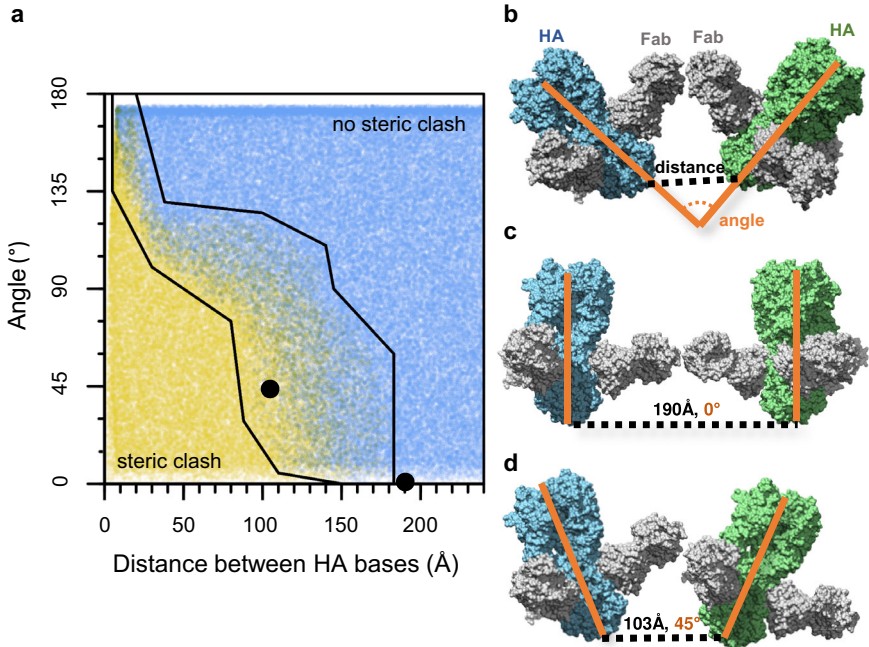

**Fig. 2 | Monte Carlo model relating HA orientation with occlusion of stem antibody binding. a** Structures of pairs of HA-trimers (green and blue), each with three stem Fabs bound in complex (gray), were randomly oriented by rigid rotation relative to each other, and then the resulting complexes were scored for steric clash. The steric clashes were described according to the distance between the base of HA trimers, and the angle between the central axis of HA trimers (orange line). The plot of steric clashes was empirically segmented into three regions: no steric clash (blue), all steric clash (yellow), and conditional steric clash depending upon the interdigitation of Fabs (mixture of blue and yellow). Within the central region of the conditional steric clash, 57% of these configurations are free of steric clash. The two black points in panel **a** represent the example complexes depicted in panels **c** and **d**. **b** A visualization of the steric clash model system denotes the distance between HA bases, and the angle between central axes of HA trimers. **c** An example configuration shows parallel HA trimers (angle = 0°) and Fab domains rotated to face each other, but the 190 Å distance between bases avoids steric clash. **d** An example of HA trimers in the conditional clash region where Fabs interdigitate, avoiding clash in some rotations, but Fabs could easily clash in other rotations of the HA trimer.

using a Fab to represent the IgG in many bound conformations, but not all. Test configurations started with the atomic coordinates of a trimeric HA ectodomain with all three stem epitope binding sites occupied by Fab. Then a second HA-trimer complex including bound Fabs was placed randomly, proximal to the first, and the configuration was assessed for steric clash (Fig. 2a). The repetitive cycle of random placement of pairs of HA-Fab complexes and their ensuing evaluation for steric clash constituted a Monte Carlo method to empirically sample the irregular parameter space relating HA distances, relative angles, and steric clash (Fig. 2b). We segmented the resulting plot of configurations of HA-Fab complexes into three regions: (i) HA-Fab complexes always clashed with each other, (ii) stem Fabs interdigitated and may or may not clash depending on the HA rotation and (iii) HA-Fabs do not clash due to sufficient space between the two HA-Fab complexes (Fig. 2b). The Monte Carlo simulation sampled 500,000 random conformations, which empirically enumerated the steric clash landscape used to define these three regions. At certain angles between HA-trimers, clashes may occur between stem-Fabs even when the inter-HA distance is comparatively large (16 nm) relative to the Fab domain size (7 nm), effectively reducing the Fab binding in those circumstances. The simulations illustrated how increased angles between the long axis of HA molecules allow full occupancy of stem epitopes at shorter distances between HA trimers (Fig. 2c vs 2d). Thus, smaller diameter complexes with greater curvature more readily exposed their full set of stem epitopes to antibody binding, partially counteracting the presence of fewer antigens on these smaller starfish complexes.

## 3D structure of HA starfish in Flublok
To determine the structural organization of starfish-like complexes, we used cryo-electron tomography of HA complexes in Flublok (Fig. 3).

Virtual slices through Flublok complexes in the 3D tomogram revealed that HA trimers radiated out from a central location in 3D space (Fig. 3a–f, Supplementary Movie 1). While the technique of negative staining had flattened the HA-starfish complexes during visualization, cryo-electron microscopy revealed that the solution structure of HA complexes indeed emanated radially outward from a central clustering of HA transmembrane domains (Fig. 3g, h). A schematic model in which trimeric HA molecules cluster into a starfish complex via their transmembrane domains is seen forming a central hydrophobic core region (Fig. 3i, grey core). Some starfish complexes were elongated rather than spherical, with HAs emanating from a cylindrical lipid core rather than a single point. The average diameter of HA complexes was approximately 25 nm. The density profiles of the constituent HA molecules were consistent with the prefusion state of HA, one which is competent for stem-epitope binding by HA-stem targeted antibodies.

HA-trimers were modeled into each density with dimensions appropriate for HA (Fig. 3g, h), which allowed a series of HA starfish complexes from cryo-electron tomography to be characterized for steric hinderance of stem epitope binding, based upon the metrics established by Monte Carlo simulation. By plotting these experimental HA-trimer configurations on the same landscape as the Monte Carlo results (Fig. 2a), the probability of stem epitope availability was estimated. For Flublok, there was a broad distribution of angles and distances between HA molecules (Fig. 3j). The likelihood that the HA complexes in Flublok permitted binding to a full complement of stem antibodies was estimated as the sum of HAs in the permissive region of the Monte Carlo simulation, plus the HAs in the conditional clash region, scaled by the fraction estimated competent for full stem antibody binding in the conditional clash region by Monte Carlo simulation (57%). For Flublok, 49% of HAs were in the permissive region and

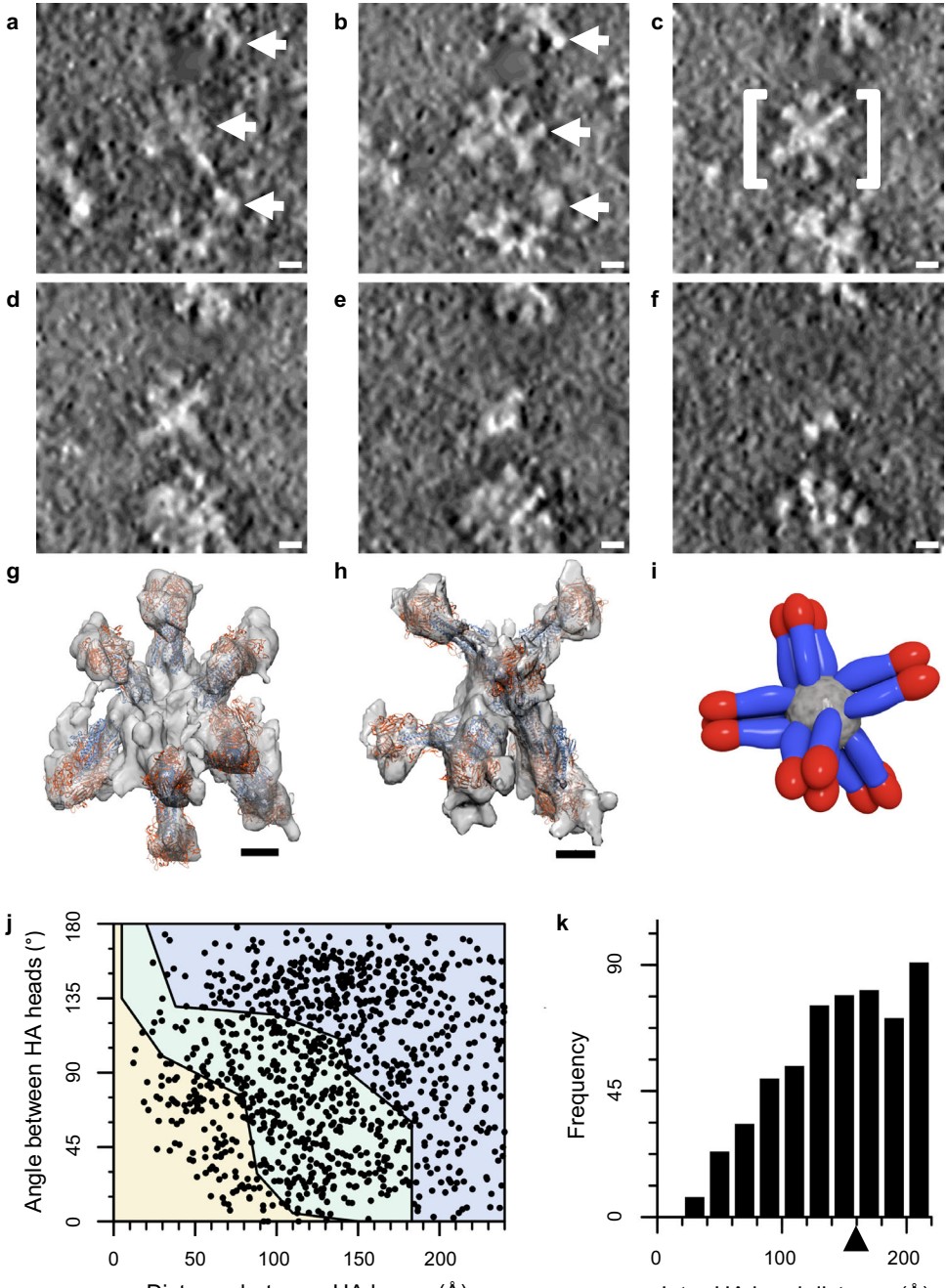

**Fig. 3 | Flublok structural analysis by cryo-electron tomography and molecular modeling. a–f** Example of six serial slices from top to bottom of a cryo-electron tomogram with a field of HA complexes. **a, b** White arrows indicate the location of three individual starfish complexes made of constituent HA molecules. **c** View in which the slice of the central complex (white brackets) presents a starfish arrangement of its HA molecules. **g–h** Two different molecular models of specific complexes with trimeric HA ectodomain coordinates, with red denoting the head and blue denoting the stem, docked into the 3D tomogram shown as a transparent isosurface. **i** A schematic of a HA-starfish structure illustrates how a central hydrophobic core (gray) contains HA transmembrane domains and lipid, and HA proteins point outward exposing the HA head colored in red, and the HA stem colored in blue. **j** Distance and angular measurements between HA-trimer bases, according to HA ectodomains docked into tomograms of the HA starfish complexes. The plot is segmented into colored regions according to Monte Carlo simulations of HA-stem clashes: no steric clash (blue), all steric clash (yellow), and conditional steric clash depending upon the interdigitation of Fabs (green). **k** HA-trimer distances between the central axis of head domains in docked complexes were tabulated and plotted as a histogram. The arrow on the x-axis denotes the threshold distance of 16 nm under which bivalent binding is possible. Scale bars are 10 nm for panels **a–f**, and 5 nm for panels **g** and **h**. Flublok cryo samples were collected at least two times and starfish morphology was consistent.

40% were in the conditionally permissive region. After scaling the fraction in the conditionally permissive region by 57%, the sum of these two regions estimates the fraction of HA stem epitopes sterically capable of binding stem antibodies in Flublok as 72%. Configuration of HAs can also impact the propensity for bivalent binding by a single antibody to two different HA heads. Bivalent binding increases the avidity of the antibody for the epitope by decreasing the off rate, enabling weak but conserved epitopes to progress through affinity maturation. The threshold for bivalent binding has been experimentally determined as approximately 16 nm[32]. Influenza vaccines that

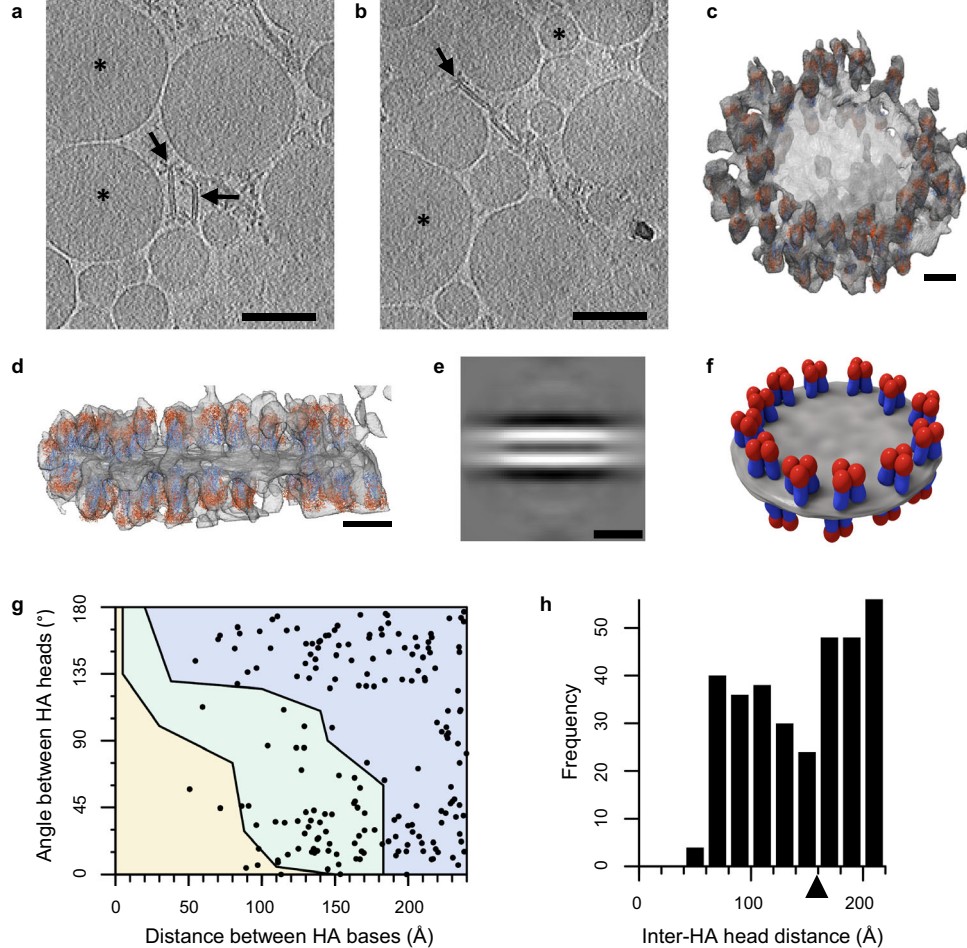

**Fig. 4 | Fluad structural analysis by cryo-electron tomography and molecular modeling. a, b** Examples of slices through cryo-electron tomograms with a field of Fluad spike complexes and putative adjuvant vesicles. Black arrows indicate some spike complexes protruding from an apparent lipid bilayer. Black asterisks denote adjuvant vesicles. **c** Oblique view of a cryoelectron-tomographic 3D reconstruction (gray) of a Fluad large particle. A disc-like structure was observed with spiked densities around its perimeter, which were used to dock HA molecules (ectodomain trimers, PDB ID 3LZG) that were parallel and anti-parallel on either the same or different sides of the disk, respectively. HA heads were colored red, and HA stems were colored blue. **d** Side-view of the Fluad particle in panel **c** illustrates parallel and anti-parallel HA orientation. **e** Side view of a subtomogram average of Fluad disc structures, focusing on the disc center. Two layers of density are visible in a density slice bisecting the disc, which indicated the discs are composed of a bilayer. The

distance between the centers of each band of density were 48 Å. **f** A schematic of the HA trimers on the perimeter of the lipid-based disc (gray) illustrates their consensus orientations with HA heads, colored in red, aligned normal to the disc surface. HA stem regions are colored blue. **g** Distance and angular measurements between HA-trimers that were docked into Fluad discs. The plot is segmented into colored regions according to Monte Carlo simulations of HA-stem clashes: no steric clash (blue), all steric clash (yellow), and conditional steric clash depending upon the interdigitation of Fabs (green). **h** HA-trimer distances between the central axis of head domains were tabulated and plotted as a histogram. The arrow denotes the threshold distance of 16 nm under which bivalent binding is possible. Scale bars are 100 nm in panels **a** and **b**, and 10 nm in panels **c**, **d**, and **e**. Fluad cryo samples were observed at least three times across multiple microscopes, with morphology of discs being consistent.

orient HA heads within 16 nm have the potential for bivalent binding, and this potential is visualized as the frequency of HA heads below this distance threshold (Fig. 3k). 63% of HA pairs in Flublok within 22 nm are within this threshold for bivalent binding.

### 3D structure of HA spiked nanodiscs in Fluad

Cryo-electron tomography of Fluad was performed to probe the 3D molecular organization and spatial display of HA stem epitopes. Fluad is a vaccine adjuvanted with MF59 (squalene oil-in-water emulsion), and we aimed to visualize the effect of the adjuvant. Virtual slices through Fluad tomograms revealed round vesicle-like structures interpreted to be the MF59 adjuvant (Fig. 4a, b, asterisks). Interspersed among the vesicles were flat disc-like structures approximately 100 nm in width, containing glycoprotein spikes (Fig. 4a, b, arrows, Supplementary Movie 2). We refer to these structures as spiked nanodiscs. HA ectodomains could be modeled into the spike densities at the

periphery of the spiked nanodiscs (Figs. 4c, d). In contrast to Flublok, the Fluad 100 nm spiked nanodiscs are larger and can contain over 50 HA trimers. To observe the nature of the lipid disc constituent in the spiked nanodiscs, we performed subtomogram averaging from 58 discs. Averaging just the center of the discs where no HA was present, we measured a distance of 48 Å between two peaks of density in a cross-section of the subtomogram averaged volume (Fig. 4e). We concluded that the spiked nanodiscs were comprised of a lipid bilayer (Fig. 4e). Because the resolution of the subtomogram average was anisotropic, we separately estimated the error of the bilayer width by manually measuring the cross-sectional distance between bilayer peaks in particles where the bilayer was viewed from the side, and we found the average and standard deviation of this distance to be 50.6 ± 4.5 Å. HA glycoprotein spikes studded the outer perimeter of the spiked nanodisc, oriented normal to the flat surface of the spike nanodisc. Thus, pairs of HA trimers were approximately parallel when

located on the same side of the spiked nanodisc, and antiparallel when on opposite sides of the spiked nanodisc (Fig. 4f, Supplementary Movie 3). This exposed the full height of HA at the edge of the spiked nanodiscs, and prominently displayed conserved stem epitopes. The fraction of HAs in the spiked nanodiscs that were permissive to full stem-antibody binding was estimated as the sum of HAs in the permissive region (67%) and the HAs in the conditionally permissive region (30%) scaled by 57% (fraction available indicated by Monte Caro simulations). Therefore, the final estimate for the fraction of HAs in Fluad permissive to full stem antibody binding was 84% (Fig. 4g). Bivalent binding, approximated by the frequency of HA heads being within 16 nm, was estimated at 62% (Fig. 4h) within the locale of 22 nm, which is notably similar to bivalent binding permissibility of Flublok starfish.

### NA quantitation

NA presence in influenza vaccines has increasingly been the focus of influenza vaccine design strategies. Because we observed individual NA tetramers in negative stain EM images, we sought to quantify bulk N1 NA content among these four influenza vaccine formulations. Focus was on N1 NA because H1N1 viruses are both seasonal and have caused pandemics in 1918 and 2009. Using western blot, we compared NA content in the vaccines to known dilutions of recombinant NA. While NA was observed by EM in all influenza vaccine formulations except Flublok, Fluad contained the highest concentration of NA (Supplementary Fig. 5). From the absolute concentration of NA, we found that the relative concentration of NA in Fluad was approximately 0.16 ug of NA for 1 ug HA, or equivalent to 2.4 ug NA per influenza strain, per vaccine dose. Meanwhile, all other vaccines contain less than our detection threshold of 0.025 ug NA per 1 ug HA (Supplementary Fig. 5).

### Induction of heterologous HA subtype antibodies

Differences in HA stem epitope displayed from structural studies of commercial influenza vaccines led us to closely investigate the in vivo immunogenicity of these same formulations. To probe these differences, we assessed binding (in vitro display) by enzyme linked immunosorbent assay (ELISA) using stem antibodies. Recombinant H1 protein was used as a positive binding control since both stem antibodies and vaccinated animal sera bind to it more readily than HA on virus (Supplementary Fig. 6). Anti-stem monoclonal antibodies FI6v3, CR6261, and C179 were used as capture antibodies to detect differences in the availability of stem epitopes in the commercial influenza vaccines Fluad, Flublok, Fluzone HD, and Flucelvax (Fig. 5a). All vaccines contained HA protein capable of binding epitopes to anti-HA stem antibodies FI6v3 and CR6261. However, while all vaccines showed similar levels of binding to FI6v3, after normalizing the formulations for HA concentration, there were differences in binding to CR6261 indicating that both Flublok and Fluad had more stem epitope binding availability than Fluzone HD and Flucelvax. These differences were more pronounced when capture antibody C179 was utilized, C179 bound significantly more Flublok than any other vaccine (Fluad, Fluzone HD, and Flucelvax). C179 was particularly poor at binding the H1 HA in Fluclevax, showing no difference from saline (Fig. 5a). These results indicated that the binding availability of conserved HA stem epitopes could vary between different commercial influenza vaccines that were purified and formulated by different methods, despite their similar composition of full-length HA trimers (Supplementary Fig. 1).

An in vivo immunogenicity study was conducted using the commercial vaccines formulations to determine if the stem epitope displays observed in vitro would be able to elicit stem antibodies in vivo. Mice were immunized with the vaccines using a prime and boost immunization schedule (Fig. 5b). Vaccine HA antigens were immunogenic and elicited homosubtypic binding antibodies to their H1, H3, and influenza B virus HA antigens, as judged by ELISAs using immunized mouse sera and full-length recombinant HA proteins

(Supplementary Fig. 7). When comparing endpoint titers calculated from dilution curves (Supplementary Fig. 8 and Supplementary Fig. 9) there were statistically significant differences between the different vaccine sera binding to H1 head, H1 stem, H2 HA and H7 HA recombinant proteins (Fig. 5c–f). Fluad sera had higher levels of antibodies binding to a H1 head domain protein than the sera from the other three vaccines (Flublok, Fluzone HD, Flucelvax) (Fig. 5c). Likewise, Fluad elicited higher levels of cross-reactive antibodies to an H1 stem protein (Fig. 5d) when compared to the other vaccines. Although H2 and H7 HA antigens were not components of the vaccines, Fluad elicited statistically significant higher levels of cross-reactive antibodies to both H2 HA and H7 HA when compared to the other vaccines, indicating the elicitation of higher levels of heterosubtypic antibodies by Fluad (Fig. 5e, f).

### Weight-loss and lung pathology differences between vaccines

Having observed differences in both the vaccines' organizations and the elicitation of cross-reactive antibodies, we challenged mice with a A/Michigan/45/2015 (H1N1) virus that was a match to the H1 vaccine component and looked for differences in survival, weight loss (Fig. 6), and lung pathology (Fig. 7). As expected, all vaccines protected mice from H1N1 challenge (Fig. 6b). This was expected because all the vaccines contained H1 HA protein and elicited antibodies to H1 HA (Supplementary Figs. 1, 2, 7). However, further examination of the weight-loss curves showed that Fluad and Flublok had trends for reduced weight loss compared to both Flucelvax and Fluzone HD (Fig. 6c). The most observable difference was at day 3 (Fig. 6c, arrow). When the experimental timeline was repeated with additional mice for the collection of lung tissue, the weight-loss differences observed on day 3 post challenge were statistically significant (Fig. 6d). Vaccination with any vaccine followed by viral challenge resulted in lesser weight loss when compared to the saline control group. Between the vaccine groups, mice given Flublok and Fluad had no significant weight-loss difference between each other after viral challenge but did have a statistically significant reduced weight-loss when compared to Fluzone HD and Flucelvax (Fig. 6d). Interestingly, when lung tissue was homogenized, infectious virus could not be recovered from the lungs of mice vaccinated with Fluad, while the other vaccines permitted viral recovery (50% tissue culture infectious dose (TCID50) assays) (Fig. 6e). Although Fluad and Flublok did not show a statistical difference in weight-loss reduction between them, they differed in their ability to permit recovery of virus from the lungs (Fig. 6e).

Lung samples were also examined by immunohistochemistry at day 3 post-viral challenge. An unbiased, blinded method was employed for analysis of histopathology of mouse lungs following H1N1 challenge (Supplementary Fig. 10). Processed samples indicated that when compared to the healthy-lung control (Fig. 7a), mice receiving saline instead of vaccine had large amounts of influenza virus NP antigen staining in lung sections (Fig. 7b, red arrows). Mice vaccinated with Fluzone HD and Flucelvax had similar distributions of viral antigen in the lungs. Animals given Flublok had lesser antigen distribution than Fluzone HD and Flucelvax. Fluad had the least amount of antigen detected in the lungs of all the vaccines tested. Fluad was visually the most similar to the healthy-lung control (Fig. 7).

### Discussion

Influenza HA is a major target of the neutralizing antibody response to influenza virus, and antibodies elicited by the immunodominant HA head domain are evaded by antigenic drift in just a few influenza seasons[33]. Antibodies targeting the HA stem are often broadly reactive across influenza virus subtypes but are elicited in low amounts by seasonal influenza virus[34]. To understand if different vaccine formulations impact the nature of the antibody response and prevalence of cross-reactive stem-binding antibodies, we evaluated 4 subunit influenza vaccines that capture the varying methods of formulating HA

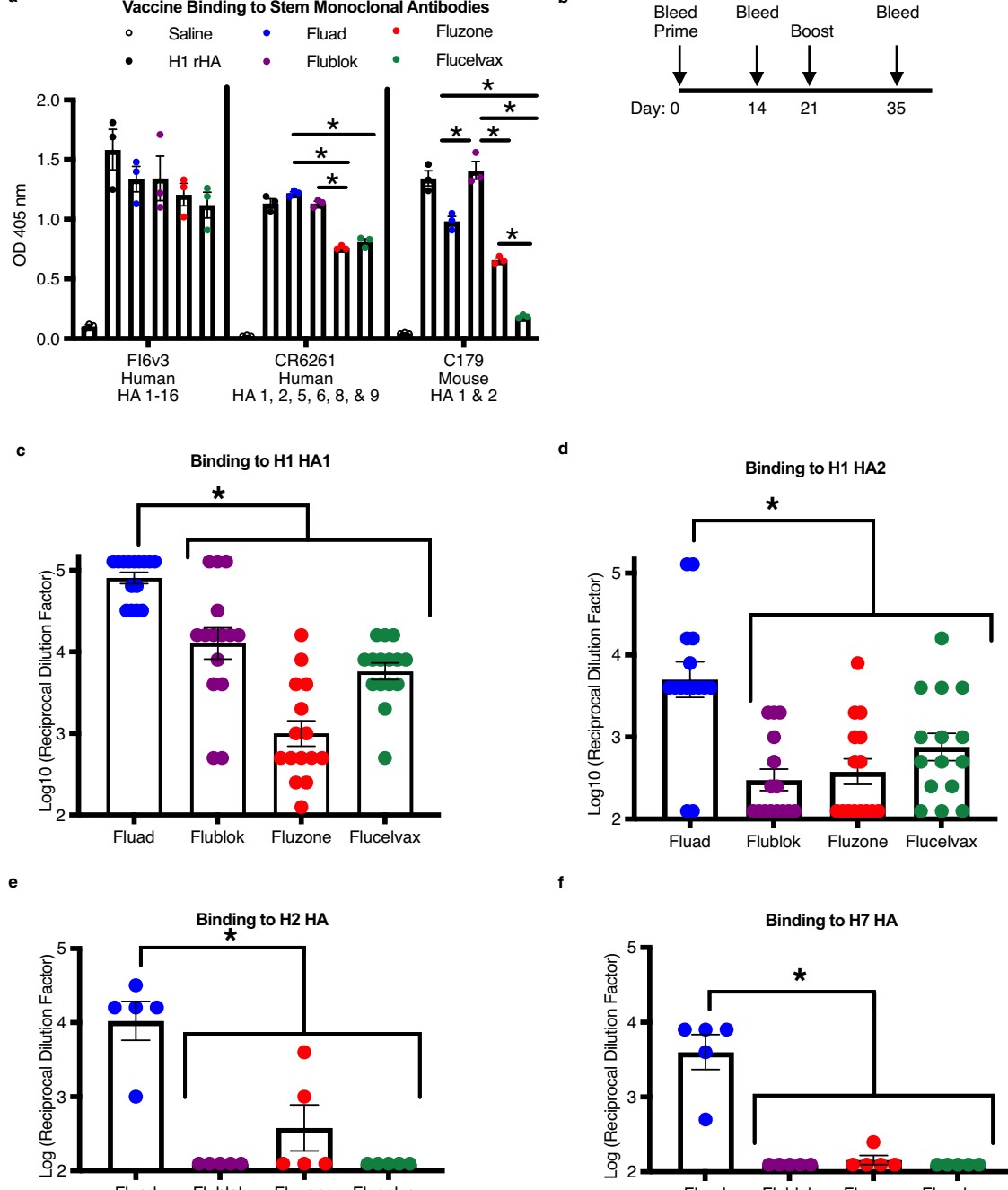

**Fig. 5 | Commercial vaccine stem-epitope availability and cross-reactive immunogenicity. a** ELISA binding analysis of commercial vaccines (Fluad (blue), Flublok (purple), Fluzone HD (red), Flucelvax (green)) by HA stem monoclonal antibodies (FI6v3, CR6261, C179). Vaccine samples were normalized for H1 HA amount prior to plating. Recombinant H1 HA protein (H1 HA A/Michigan/45/21015, black dots) was a positive control with saline (white dots) as the negative control (F (10, 36) = 7.219, p < 0.0001). **b** Schedule for immunization of mice with vaccines with day 35 sera used in subsequent ELISA studies. **c**–**f** The endpoint titer levels of different vaccine sera were compared for cross-reactivity to different recombinant proteins: **c** H1 head (N = 15 mice, F (3, 56) = 32.86, p < 0.0001), **d** H1 stem (N = 15 mice, F (3, 56) = 10.77, p < 0.0001), **e** H2 HA (N = 5 mice, F (3, 16) = 20.16 p < 0.0001), and **f** H7 HA (N = 5 mice, F (3, 16) = 38.08, p < 0.0001). PBS control sera were also tested to determine threshold, but not shown. Graphs display the mean with standard error of the mean bars and asterisks to denote statistical differences (p < 0.05, Tukey's multiple comparisons test with two-tailed significance).

for vaccination (i.e., Fluzone HD, Flucelvax, Flublok, Fluad) (Fig. 1, Supplementary Fig. 1). Recent studies using structure-guided nano-particle designs for viral vaccines have indicated numerous important factors for glycoprotein display and immunogenicity. These factors may include glycoprotein pre/post-fusion conformation, glycoprotein spacing, and number of glycoproteins or multivalency of the nanoparticle[28,35,36]. However, these factors such as HA conformation,

spacing and numbers have not been studied in terms of current influenza commercial vaccines in a side-by-side comparison. Other types of antigen display are currently being developed to enhance immunogenicity of vaccines, such as octahedral ferritin particles, to display the prefusion trimeric ectodomains of a multitude of different viral proteins such as influenza HA, human immunodeficiency virus 1 (HIV-1) Env, and severe acute respiratory syndrome coronavirus 2

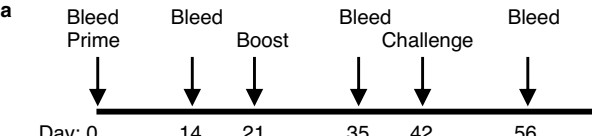

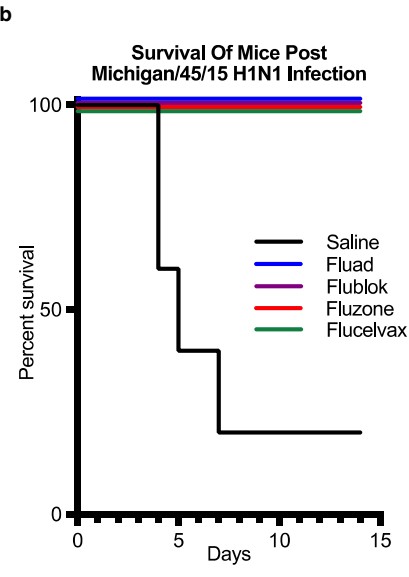

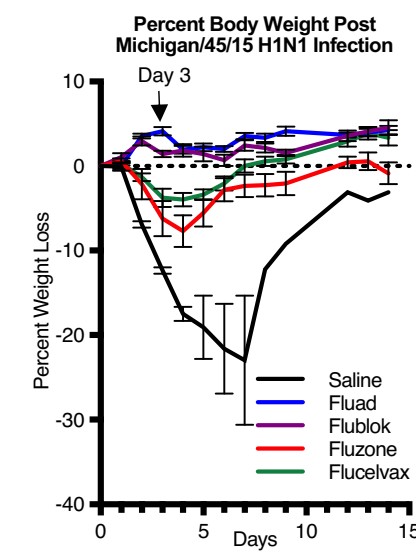

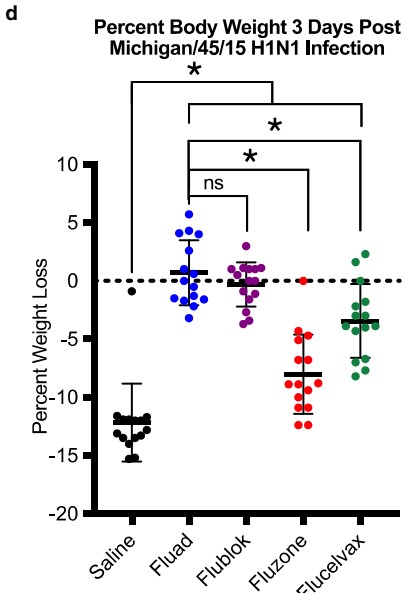

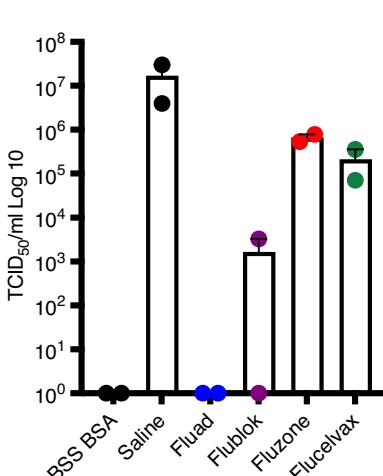

**Fig. 6 | Protection efficacy and weight lost differences after H1N1 challenge.**
**a** Mice were immunized with commercial influenza vaccines (day 0 and 21) and challenge with A/Michigan/45/2015 H1N1 influenza virus at 10x the lethal dose for control mice (day 42). **b** Survival curves for mice ($N = 5$) immunized with saline control (black), Fluad (blue), Flublok (purple), Fluzone HD (red) and Flucelvax (green). **c** Weight-loss curves for challenged mice ($N = 5$) that were immunized with saline (black), Fluad (blue), Flublok (purple), Fluzone HD (red) and Flucelvax (green). The black arrow indicates weight-loss on day three post H1N1 challenge. **d** Statistical analysis of weight-loss differences ($N = 15$ mice) between immunization groups for day 3 post H1N1 challenge (F (4, 70) = 49.44, $p < 0.0001$). Tukey's

multiple comparisons test with two-tailed significance found no statistical difference in weight-loss between Flublok and Fluad as denoted (i.e., ns.). Asterisks denote statistical differences ($p < 0.05$) in weight-loss between saline control as compared to vaccines (Flublok, Fluad, Fluzone HD, Flucelvax) and statistical differences between Flublok and Fluad versus the other vaccines, Fluzone HD and Flucelvax. **e** Mouse lung titers were determined by 50% Tissue Culture infectious Dose (TCID50) for tissues taken day three post-H1N1 challenge and compared between commercial influenza vaccines. Graphs display mean with standard error of the mean bars.

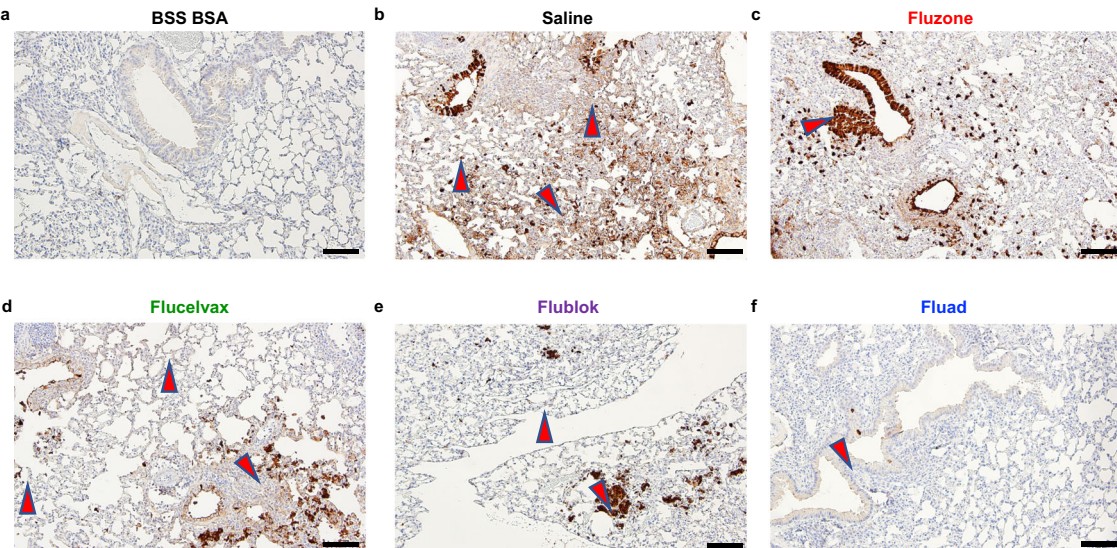

**Fig. 7 | Histopathology in mice lungs following vaccination with commercial influenza vaccines and subsequent challenge with H1N1 influenza virus.** **a**–**f** Images show immunohistochemistry (IHC) against the influenza nucleoprotein protein for different vaccination groups. Red arrows denote viral antigen areas. Images shown are from day 3 post H1N1 challenge for all groups. **a** Healthy lung control via immunization with saline and challenged with saline-BSA. **b** Pathogenic positive control via immunization with saline with challenge with H1N1 virus. **c**–**f** Immunization with the commercial influenza vaccines **c** Fluzone HD, **d** Flucelvax, **e** Flublok, and **f** Fluad and subsequent challenge with H1N1 virus. Lung samples were collected from two mice per condition group and the experiment was replicated (N = 4). All images at 10x magnification and display 100 µm scale bars.

(SARS-CoV-2) spike[28,36–38]. Although commercial vaccines are given to millions of people world-wide, differences in stem epitope display and vaccine structure between a group of influenza vaccines generally have not been studied. Previous reports have been limited to biochemical or human/animal studies for individual commercial influenza vaccines[21,39].

In this work, we directly observe a variety of parameters that describe how epitopes are made available in commercial influenza vaccines. We started by quantitating epitope bioavailability via monoclonal antibody binding. By normalizing the concentration of H1 HA protein, we found that the Flublok vaccine had the highest fraction of H1 HA stem bound across the monoclonal stem antibodies examined, while Fluzone HD, Flucelvax and Fluad formulations varied in binding based upon the selected capture antibody (Fig. 5a). We also found differences in the induction of stem-targeted antibodies in mice, which did not follow the same trend as available stem epitopes in vaccine formulations (Fig. 5d–f). While Flublok exposed the most stem epitopes in vitro, it induced less HA stem-specific antibodies in mice. Meanwhile, Fluad which exposed similar or lower levels of stem epitopes in vitro as Flublok, elicited significantly more HA stem-specific antibodies in mice. Fluad also elicited the most HA head-specific antibodies, conferring it the strongest HA-specific antibody response overall. Fluzone HD contained the most non-HA viral components and offered the most heterogenous configurations of HA clusters, while inducing the least amount of head-specific HA antibodies.

We found that HA complexes in commercial influenza vaccines did not exhibit any regular symmetry (Figs. 1, 3, 4). Different from HA starfish-like complexes found in the engineered HA ferritin nanoparticles[28,36], commercial vaccines present a medley of arrangements for a given number of HAs, which are made possible by full-length HAs being embedded in a lipid component. Our structural studies suggests that in the vaccines, the HA molecules are in a pre-fusion state and that the mechanism of HA complex formation appears to be via aggregation of the hydrophobic transmembrane (TM) regions (Fig. 3g, h).

Molecular models from 3D cryo-electron tomography indicate that in Flublok, starfish-like structures are bound together by hydrophobic TM regions which partition out of aqueous solution (Fig. 3a–i). In contrast, HA trimers in Fluad were arranged as a double ring with a disc shape that we refer to as a spiked nanodiscs (Fig. 4). This nanodisc structure is a different geometrical display of HA molecules than both the HA starfish arrangement and rigid nanoparticles, such as those scaffolded by ferritin[28,36]. The Fluad spiked-nanodisc structure could be a novel method of displaying viral gly-coproteins that is optimized to present conserved stem epitopes. These larger spiked nanodiscs also contain more HA trimers than most nanoparticle scaffolds, perhaps equivalent to icosahedrally symmetric vehicles. Compared to other subunit vaccines, Fluad has higher mul-tivalency of HA complexes (Fig. 4c vs 3 g). While Fluad complexes would have less multivalency than whole virus or virosomes, the par-ticle concentration of these virus-sized complexes would also be lower and would require direct comparison to evaluate.

Ideal spacing and presentation of antigen for immune stimulation has long been an interest when rationally designing vaccines[35]. Densely spaced antigens on a particle surface offer more binding sites to interact with immune system components, yet very densely packed antigens can occlude access to epitopes recessed on the antigen sur-face. IgG is comprised of three rigid domains: two Fab domains which each separately bind antigen, and an Fc domain that binds receptors and immune effectors. Bivalent binding by IgG can occur when epi-topes are spaced between 4–16 nm apart[32], resulting in high avidity binding. When antigens are densely spaced but not compatible with bivalent finding, Fc and Fab domains may sterically hinder IgG from binding neighboring antigens. The linkers between the three domains in IgG have preferred orientations, but are flexible to a degree that varies by antibody subtype[31]. Since the bound Fab domain is the only fragment of the IgG rigidly positioned by antigen binding, we employed a Monte Carlo method to directly evaluate the irregular molecular surfaces of HA trimers bound to stem-targeting Fab. We found that HA stem epitopes are readily occluded when HA molecules pack closely next to each other, yet stem epitopes become more available as the angle between HA trimers increases (Fig. 2). Thus, there is an interesting relationship between multivalency and epitope accessibility, as increased density of HA antigen can both occlude

epitopes and offer more opportunity for bivalent binding. Meanwhile, smaller complexes with lower multivalency gain epitope accessibility through less HA being present, as well as greater particle curvature.

Our simulations address only the rigid Fab domain, but the impact of full IgG or even IgM should also be considered. In supplementary work (Supplementary Fig. 11) we first used the 5 crystal structures of full-length IgG (PDB IDs 1HZH, 1IGT, 1IGY, 5DK3, 6GFE), which we aligned to representative poses of HA trimers illustrated in Fig. 2. When HA trimers were positioned at a 90 degree angle (Fig. 2b), we were surprised to find that the off-center attachment site of the Fc-Fab linker as it connects to the Fab domain often avoided steric clashes that were anticipated due to close proximity of bound Fab domains. As HA trimer orientations became parallel and the orientation of bound Fab domains approached 180 degrees, clashes were observed with either the Fc domain or the unbound Fab domain (Supplementary Fig. 11). These instances, where Fab domains were just beyond the range of bivalent binding yet still oriented in opposite directions, were the conformations most affected by the presence of the full-length IgG. Still, quantitation of this effect would be specific to antibody type and subtype, and would require rigorous knowledge of the linker length and flexibility, such as described for IgG1[31] and IgD[40]. In an intriguing case where IgG structure was resolved in bivalently-bound complexes of 4-, 5-, and 6-membered rings with its CD20 antigen, Fc domains were poorly resolved due to substantial flexibility even when both Fab domains were immobilized by antigen[41]. We conclude that while the physical presence of Fc and unbound Fab domains must play a role in the sterics of binding dense antigen, the significance is minimal for most binding orientations of this HA model system due to the Fc-Fab linker positioning and flexibility.

Previous studies with viral glycoproteins, such as Epstein-Barr virus gp350 and influenza HA trimer, indicated that increasing multivalency via a ferritin nanoparticle yielded increased immunogenicity and vaccine efficacy in animal models[36,42]. For influenza commercial vaccines, our work suggests that low multivalency could have HA stem in vitro binding activity by anti-HA stem monoclonals like for Fluzone HD (Fig. 5a red dots), but not elicit higher levels of antibodies to the stem or cross-reactive ones to different HA subtypes in vivo (Fig. 5d–f). Fluzone HD contained HA starfish-like complexes of relatively low numbers of HA (i.e. low multivalency) (Fig. 1a, b). In contrast, Fluad had similar levels of in vitro binding by the human anti-stem broadly reactive antibody FI6v3 (Fig. 5a blue dots) yet elicited the highest levels of cross-reactive antibodies to the stem and to HA subtypes not a constituent in the vaccines (Fig. 5d–f). This ability of Fluad to elicit cross-reactive antibodies coincided with the observation by cryo-electron microscopy of the unique spiked nanodiscs, containing a variable number of HAs commonly over 50. The spiked-nanodisc structure is unique in how it displays parallel HA trimers, exposing stem epitopes around the disc's perimeter in such a way as to promote multivalent binding specifically for stem epitopes. In creating these spiked nanodisc structures, the adjuvant may act to directly arrange the HA trimers and improve the elicitation of cross-reactive antibodies.

Dominance of the immune response to highly variable epitopes on the HA head is a known challenge in influenza vaccine design, and is an important factor in limiting the effectiveness of immunity due to natural infection[14]. Whether or not HA stem epitopes are selected against in B cell maturation due to the nature of the HA protein itself is an ongoing question. While not outright solving this problem, we demonstrate in this work that even among full-length HA antigens that include the variable head epitopes, improvements can be made in targeting a greater fraction of the immune response to conserved stem epitopes (Fig. 5). Even though designed antigens may illicit only a stem epitope response, antibodies targeting the head typically neutralize virus much more effectively when the viral strain is matched to the vaccine[14]. Thus, it is desirable to understand how a single vaccine may elicit both head and stem antibodies in a way that neither response

dominates over the other. Here, we have both described how four influenza vaccines differ in their organization structure of antigen, as well as the nature of their response, hopefully pointing the way for future vaccines to build upon the best of these characteristics.

In conclusion, although all of the commercial influenza vaccines in this study contained full-length HA proteins there were differences in the number of HA molecules per particle (multivalency), spacing, and epitope display within HA complexes. Our results suggest that vaccine formulations, like Fluad, that specifically structure HA antigen into spiked-nanodisc displays can offer examples of how to improve multivalency and immunogenicity of HA subunit vaccines to formulate more efficacious influenza vaccines for pandemic preparedness and universal influenza vaccine development. More broadly, these structure-guided approaches to influenza subunit vaccines containing HA can be applied to viral glycoproteins from other viruses with pandemic potential such as HIV, Ebola, and SARS coronaviruses.

## Methods

### Negative-stain electron microscopy (EM)
Samples (e.g. Flucelvax, Fluzone HD, Flublok, and Fluad) were diluted in phosphate-buffered saline (PBS) to an HA concentration of 10 μg/mL. EM sample grids were prepared by applying 5 μL of the sample to glow-discharged Formvar/carbon (F/C) 300 mesh grids. Excess sample was blotted with filter paper, then the sample was rinsed twice by dipping in deionized water and blotting with filter paper. The specimens were negatively stained by dipping the grid in 3% phosphotungstic acid at pH 7 for 1 min, blotted, rinsed one time by deionized water, blotted dry with filter paper, and then dried[43]. Data was acquired using a Tecnai 12 electron microscope (ThermoFisher) with a LaB$_6$ filament operating at 100 kV and nominal magnifications of 67,000x, yielding a magnified pixel size of 1.8 Å/pixel. All images were recorded on a 4kx4k OneView camera (Gatan, Pleasaton, CA) using SerialEM software. Data for class averages was typically acquired in a single session lasting 1 to 2 hours, which yielded 100-200 micrographs[44]. For 2018–2019 Flucelvax, 134 micrographs (12,652 particles) were collected; for 2018–2019 Fluzone HD, 136 micrographs (15,739 particles) were collected; for 2018-2019 Flublok, 27 micrographs (4575 particles) were collected; and for 2018–2019 Fluad, 97 micrographs (949 particles) were collected.

Negative stain EM data were processed using RELION3 v.3.1.2[45]. The contrast transfer function (CTF) for each micrograph was estimated using CTFFIND4 software[46]. For 2D image averaging of whole vaccine particles, for each vaccine, 500-1,000 particles were manually picked. RELION3 2D classification was then used to generate autopicking templates, which were used as references to repick particles on all micrographs, yielding a total of 10,000 to 20,000 particles that were extracted at a box size of 256 pixels. These particles were aligned and classified using RELION3 2D Classification to remove poorly resolved classes (e.g. false positive picks or poorly aligned particles) from the particle set. Results from the final round of 2D classification were used to characterize the sample.

For 2D image averaging of individual HA proteins within vaccine particles, only manually picked particles were used to ensure HA proteins were isolated and proximal particles didn't interfere with averages. 2017–2018 Flucelvax 176 particles were manually picked. For 2017–2018 Fluad, 135 particles were manually picked. For 2017–2018 Fluzone HD, 143 particles were manually picked. For 2022 Flublok, 152 particles were manually picked. At a resolution of 1.8 Å/pixel, particles from Fluad and Flucelvax were extracted at a box size of 176 pixels while particles from Fluzone HD and Flublok were extracted at a box size of 128 pixels. The smaller box sizes were used to avoid nearby particles. These particles were aligned and classified using RELION3 2D classification with a single class. Classifications with 2, 3, or 4 classes were also investigated, but results did not offer further insight. Fiji

software[47] was used to measure HA particle height in 2D class averages, and to produce an intensity profile along the height of the HA particle. Final class averages were manually rotated to vertically align HA to facilitate comparison across vaccines. Images of HA reference structures were generated using Chimera[48] and 2D density projections from 3D volumes were made with EMAN2 v.2.91[49].

## Monte Carlo simulation

The crystal structure of HA in complex with stem Fab C179, PDB code 4HLZ, was used to model steric clashes between stem antibodies when bound to HA. Using the PERL scripting language, coordinates for the complex were rotated a random amount, then translated a random amount up to 25 nm and written to a PDB file including the rotated HA complex and a reference HA complex. The same wrapper PERL script then calls Chimera[48] to measure distances and angles between these two HA complexes, as specified by a Chimera command file. Steric clashes between these two HA complexes were scored using the software MolProbability Probe[50]. A cycle consisting of model generation, measurement, and clash scoring constituted one iteration of the Monte Carlo Method, and 500,000 such iterations were run. Plots were generated using the statistical software R[51].

## Cryo-electron tomography

Cryo-electron microscopy grids for 2018-2019 Flublok and 2018-2019 Fluad were prepared separately using similar procedures. Samples were mixed with 5 or 10 nm gold particles as fiducials. Using a Leica EM GP plunge freezer (Leica Microsystems), aliquots of 4 µL of sample were applied to glow-discharged, Quantifoil C Flat R2/2 200 mesh grids, blotted for 1.5 seconds in 99% humidity chamber, and plunge-frozen in liquid ethane. Flublok and Fluad tilt series were collected using Falcon II and Falcon 3 cameras on Titan Krios microscopes operating at 300 kV using a Volta phase plate (VPP) (FEI). Zero loss energy filtered data was collected with a 70 µm objective aperture at 3.3 Å pixel size (nominal magnification 26,000x) with a defocus range of −3.5 to −5.5 µm. Additionally, Fluad tilt series data was collected with a Titan Krios microscope (ThermoFisher) operating at 300 kV but with a GIF-2002 energy filter (Gatan Inc.) with a slit width of 20 eV, coupled with a Gatan K2 camera (Gatan Inc.), 2.9 Å pixel size (nominal magnification 29,000x) and a − 2.5 µm defocus. All single-axis tilt series data were collected over − /+ 60° with 2° tilt increments. Tilt series datasets were collected under low-dose conditions with a cumulative electron dose of 100 e⁻/Å². Images within the tilt series were mutually aligned by using the gold particles as fiducial markers and the 3D volumes (tomograms) were reconstructed as implemented the IMOD v.4.10.16 and EMAN2 v.2.91 software packages[52,53].

Docking of HA trimers into subtomograms was performed using Chimera[48]. A total of 20 subtomograms of Flublok and 7 subtomograms of Fluad were analyzed. Scripted commands in chimera were used to measure distances and angles between HA trimers. PERL scripts were used to parse data in preparation for plotting, and R was used to generate plots.

For subtomogram averages of Fluad discs, particles were picked from tomograms that had been reconstructed with EMAN2[53]. Particles were extracted and subtomogram averaged using RELION2 v.2.1[54]. To resolve the central lipid comprising the discs, particle box sizes was chosen to be 64 pixels, which was 346 Å at a working pixel size of 5.4 Å/pixel. To arrive at a single best-fit measurement of the disc thickness, D64 symmetry was imposed. 58 nanodiscs were used in the final subtomogram average structure.

## Western blot for vaccine composition and NA quantitation

Denatured N1 NA protein c26 Cal09 WT was provided by the Vaccine Research Center (NIH, Bethesda, MD) and loaded onto an SDS-PAGE gel (4-12% Bis-Tris) at twofold dilutions beginning at 120 ng. Denatured

vaccines (Fluad, Flublok, Fluzone HD, Flucelvax) and recombinant H1 HA A/Michigan/45/15 (catalogue number 3006, Protein Sciences) were loaded at an amount of 0.3 µg. Gel electrophoresis was run at 200 V for 50 min to separate vaccine components. Bands were transferred to nitrocellulose membrane via the iblot transfer system (Thermofisher) and probed with primary antibody and florescent tagged secondary antibody via lateral flow (ibind system, Thermofisher). For vaccine antigen composition western blots, primary antibodies were anti-H1 polyclonal PS6000 (Protein Sciences, 1:1000 dilution), anti-H3 polyclonal PS6001, (Protein Sciences, 1:1000 dilution), anti-HB polyclonal PS6003 (Protein Sciences, 1:1000 dilution), anti-NA polyclonal PA5-32238 (GeneTex, 1:1000 dilution), anti-M1 monoclonal HB64 (American Type Culture Collection clone M2-1C6-4R3, 3.5 µg/ml), and anti-NP monoclonal HB65 (American Type Culture Collection clone H16-L10-4R5, 3.5 µg/ml). For NA quantitation, NA antibody PA5-32238 (GeneTex, 1:1000 dilution) was used. Membranes were visualized on the c600 imager (Azure biosystems) for 1 sec and 30 sec[47]. NA standard curve was generated using Prism (GraphPad Software) and the NA concentration for the vaccines was extrapolated. Blots were done in duplicate to ensure accuracy.

## Propagation and titration of H1 virus

Madin-Darby Canine Kidney (MDCK) Cells, London Line (FR-58) was obtained through the International Reagent Resource (IRR). These cells were grown in DMEM (Gibco) supplemented with FBS 10%, Pen Strep (Gibco). Cells were infected with A/Michigan/45/2015(H1N1) pdm09 (IRR, FR-1483), at 37 °C/72 hr, using DMEM without addition of FBS or antibiotic. Then, the cell culture supernatants were obtained by centrifugation at 1800 x g, at 4°C, for 20 min to remove cell debris. Samples were concentrated by ultracentrifugation at 50,000 x g, 8 °C for 2 h, and the pellet was resuspended in 500 µl of Phosphate-Buffered Saline (PBS). The virus inoculum was back-titrated to verify the accuracy of the hemagglutination activity units (HAU) to 64 HAU/ 50 µl. Fifty microliters volume of virus was added in the first well, then 2-fold serial dilutions in V-well microtiter plates were incubated with 0.75% turkey RBCs for 30 min at RT. After incubation was complete, individual plates were placed in the Cypher One instrument (InDevR) for imaging.

## ELISA for epitope accessibility

Stem monoclonal antibodies (FI6v3, CR6261, and C179 provided by the Vaccine Research Center, NIH, Bethesda, MD) were applied to microtiter plates (immulon 2 B, Thermo Fisher) at 1.25 µg/ml to act as a capture antigen and incubated overnight at 4 °C. Plates were washed between all experimental step with PBS plus 0.1% tween 20 (405 TS ELISA plate washer, Agilent Technologies). Blocking (1% omniblok, AmericanBio, Inc., and 0.1% tween 20 in PBS) occurred for 30 min at 20 °C prior to application of commercial vaccines or rHA at 1.25 µg/ml of H1 HA or saline for 2 h before washing and probing with a rabbit polyclonal antibody to H1 HA (PS6000, Protein Sciences, 1:1000 dilution) for an additional 2 h. Plates were than incubated at 37°C for 1 h with a goat anti-rabbit horseradish peroxidase (HRP) conjugated secondary antibody (goat anti-rabbit IgG (H + L), Thermo Fisher). Colorimetric detection occurred for 15 min at 20 °C and absorbance was read at 405 nm (1-Step 2,2'azinobis-[3-ethylbenzothiazoline-6-sulfonic acid] diammonium salt (ABTS), Thermo Fisher) at 405 nm. Samples were run in triplicate. A background control was subtracted and the data is displayed as mean with standard error of the mean (SEM) bars using Prism (GraphPad).

For determination of mouse sera immunogenicity, antigen was applied to 96-well plates and incubated overnight at 4 °C (1.25 µg/mL). Recombinant H1 head antigen was obtained from the International Reagent Resource (IRR), catalog number FR695. Recombinant H1 stem antigen comprised mostly of HA2 sequence only was kindly provided by Jeffery Taubenberger from the Laboratory of Infectious Diseases,

NIH NIAID. Recombinant H2 antigen was obtained from IRR, catalog number FR700. Additional HA antigen was obtained from Protein Sciences (H7 A/Anhui/01/2013, H1 A/Michigan/45/2015, H3 A/HongKong/1408/2014, and B Victoria B/Brisbane/60/2008). Plates followed a similar procedure as above for washes, blocking, incubations and detection. Primary serum samples from vaccinated mice were plated at an 1:125 dilution with two-fold dilutions following. Endpoint titer levels were statistically defined per plate[55] as the lowest sera dilution still differing from the saline group, an endpoint titer of 2.01 Log(reciprocal dilution factor) indicates no difference from the saline group at the highest sera concentration tested. Statistical analysis was conducted for analysis of variance (ANOVA) using Prism (GraphPad), statistically significant differences are denoted with an asterisk to represent a significance level (p) less than 0.05.

To compare binding between rHA and HA displayed on virus, ELISA plates were incubated overnight at 4°C with rHA (2 μg/ml) or virus (ranging from 8 to 64 HAU/50 μl[56]). Plates were washed and blocked prior to application of primary antibody: either monoclonal stem antibodies (10 μg/ml) or sera (1:125 dilution). Finally, plates were detected with the appropriate HRP conjugated secondary and ABTS. Samples were run in duplicate. A background control was subtracted and the data is displayed as mean with standard error of the mean (SEM) bars using Prism (GraphPad).

### Animal immunization and mouse challenge

Rabbit studies were conducted by Life Sciences Solutions (Thermo Fisher Scientific) and were in accordance with the Animal Care and Use Committee (ACUC) of Cocalico Biologicals, Inc. New Zealand White rabbits (Cocalico Biologicals) began experimentation at approximately two to four months of age. Polyclonal rabbit sera were collected for 1–2 male rabbits per vaccine (Fluad, Flublok, Fluzone HD, or Flucelvax). Rabbits were injected intramuscularly with a total of 1 ml of vaccine on days 0, 14, and 42 with blood collection occurring immediately prior to immunization and final blood collection occurring on day 56 (Supplementary Fig. 6c).

All mouse experiments were performed under protocols approved by the ACUC at the National Institute of Allergy and Infectious Diseases (NIAID). Mouse immunogenicity studies were conducted with 25 female BALB/c mice (Taconic Biosciences) aged 8–10 weeks that were randomly assigned to a vaccine group. Female mice were utilized for in vivo experimentation due to their higher immunogenetic response following vaccination[57–59]. Groups of mice (N = 5) received one of five types of intramuscular injections for immunization on day 0 (prime) and day 21 (boost): saline, Fluad, Flublok, Fluzone HD, and Flucelvax. Mice were injected intramuscularly (50 μL/leg) at the manufacturer's supplied concentration. Mice were weighed weekly to ensure health with bleeds occurring on day 0, day 14, and day 35 to track immunogenicity.

On day 42 mice were anesthetized and intranasal challenged with 25 μl of influenza virus H1N1 A/Michigan/45/2015 at $10 \times MLD_{50}$ (50% Mouse Lethal Dose). The initial survival experiments used five mice per vaccine group and were observed twice daily for survival criteria (humane endpoints observed following a 30% decrease from initial body weight) until day 56, when all surviving mice were humanely euthanized. Differences in survival rates were compared using a Kaplan–Meier survival analysis (Graph Pad Prism). A replicate study was conducted with mice being euthanized 3 days post challenge for the collection of lung tissue for pathology. A control mouse group for pathology studies received vaccinations with saline on day 0 and day 21, and challenge with saline supplemented with 0.1% bovine serum albumin (BSS-BSA) on day 42.

### Determination of influenza virus mouse lung titer

TCID50 assays were conducted by infecting the ATL isolate of Madin-Darby canine kidney cells (MDCK-ATL) cells with 1:5 serial dilutions of each mouse lung homogenate in duplicate in serum-free minimum essential media (MEM) with 1 μl/ml N-p-tosyl-L-phenylalanine chloromethyl ketone (TPCK)-treated trypsin and incubated at 37 °C and 5% $CO_2$ for 96 h. Cytopathic effect (CPE) was determined following incubation by staining with crystal violet and TCID50 was calculated using the Reed-Muench method[60].

### Immunohistochemistry

Formalin-fixed paraffin-embedded mouse lung sections were labeled for immunohistochemical staining using a rabbit polyclonal anti-Influenza A virus nucleoprotein antibody, CAT# GTX125989 (Gene-Tex). Staining was carried out on the Bond RX (Leica Biosystems) platform according to manufacturer-supplied protocols. Briefly, 5 μm-thick sections were deparaffinized and rehydrated. Heat-induced epitope retrieval (HIER) was performed using Epitope Retrieval Solution 2, pH 9.0, heated to 100 °C for 20 minutes. The specimen was then incubated with hydrogen peroxide to quench endogenous peroxidase activity prior to applying the primary antibody at a dilution of 1:4000. Detection with 3,3'-diaminobenzidine (DAB) chromogen was completed using the Bond Polymer Refine Detection kit (Leica Biosystems). Slides were finally cleared through gradient alcohol and xylene washes prior to mounting and coverslipping. Sections were examined by light microscopy using an Olympus BX51 microscope and photomicrographs were taken using an Olympus DP73 camera. An unbiased blinded then unblinded approach was used to analyze the immunization conditions of mice using commercial influenza vaccines with histopathology after H1N1 viral challenge. Pathology experiments were repeated to ensure accuracy, N = 4.

### Reporting summary

Further information on research design is available in the Nature Portfolio Reporting Summary linked to this article.

### Data availability

The source data underlying Figs. 2a, 3j, 4g, 5a,c,d,e,f and 6b,c,d,e and Supplementary Figs. 4i,j, 5c, and 6a,b,d are provided in SourceData.xlsx inside SourceData.zip. Tomographic reconstructions have been deposited in the Electron Microscopy Data Bank, https://www.ebi.ac.uk/emdb/, under the following accession numbers: Fluad spiked nanodisc, "EMDB-27232"; Flublok starfish, "EMD-27233". PDB files used in this study were downloaded from the Protein Data Bank: https://www.rcsb.org/, including "1HZH", "1IGT", "1IGY", "5DK3", "6GFE", "4HLZ". Due to limited available volumes, mouse sera generated in this study are unable to be made generally available. Source data are provided with this paper.

### Code availability

PERL and R scripts to reproduce Fig. 2a are included in the supplementary data file SourceData.zip.

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

## Acknowledgements

This work was supported by the Intramural Research Program of the National Institute of Allergy and Infectious Diseases, 1ZIAAI001180, AKH. We thank Jeffery Taubenberger and Jae-keun Park for kindly providing the headless trimeric stem protein and the International Reagent Resource for HA proteins. We thank Masaru Kanekiyo and Barney Graham for providing stem antibodies. We also thank Vinod Nair, Cindi Schwartz, Elizabeth Fischer, and Rick Huang for help in cryo-EM data collection. This work utilized the computational resources of the NIH HPC Biowulf cluster (http://hpc.nih.gov), and the Office of Cyber Infrastructure and Computational Biology (OCICB) High-Performance Computing (HPC) cluster at the National Institute of Allergy and Infectious Diseases (NIAID), Bethesda, MD.

## Author contributions

M.M. designed the in vivo experimental studies, ELISA experiments and western blots. J.G. designed the EM experiments and Monte Carlo simulations. A.K., W.P., H.A. and J.G. collected and analyzed EM data. S.M.-P. and U.T. grew viral reagents and conducted titer experiments. A.H., K.B., and I.M. conducted and interpreted the pathology experiments. M.M., J.G. and A.H. designed the study and experiments and wrote the paper. All authors contributed to editing the paper. All authors read and approved the final manuscript.

## Funding

## Competing interests

The authors declare no competing interests defined as financial and non-financial interests (including but not limited to funding, employment, stocks, shares, patents, personal or professional relationships with individuals or institutions, and unpaid membership advocacy) that could be perceived to directly undermine the objectivity, integrity, and value of a publication, or could be seen as having an influence on the judgments and actions of authors with regard to objective data presentation, analysis, and interpretation.
