## [Peer Review File · Nature Communications]

Reviewers' Comments:

Reviewer #1:

Remarks to the Author:

The manuscript by Myers et al presents a novel examination of the various structural arrangements of Hemagglutinin (HA) within four commercial influenza vaccines. The authors demonstrate using cryo-electron tomography and Monte Carlo simulations with HA/stem-Fab complexes that the different HA structural presentations observed within the four vaccines lead to differences in predicted antibody binding availability of conserved HA stem epitopes. Logically one would expect that increased antibody binding availability of HA stem epitopes within vaccine formulations would lead to induction of higher levels of anti-stem antibodies, which might provide greater protection for future influenza variants. Surprisingly, the authors found in mice that induction of stem-targeted antibodies did not follow the same trend as stem epitope availability within the vaccine formulations. More importantly however, the vaccine formulation with the most exposed conserved stem epitopes, as presented within a "spiked nanodiscs", elicited the highest levels of heterologous HA subtype antibodies. This result, if confirmed by further studies with vaccines for other pathogens, would be helpful in generation of vaccines that induce broader immunity. This would be incredibly helpful in the design of a universal flu vaccine or a more broadly effective coronavirus vaccine.

The methodology presented is sound and the cryo-electron tomography structural analysis is state-of-the-art. There appears to be enough detail provided in the methods for the work to be reproduced. The major strength of the manuscript is that authors demonstrate (I believe for the first time) that the structural arrangement and presentations of epitopes within a vaccine formulation is a factor that can now be directly observed and correlated with vaccine effectiveness.

Minor points and suggestions:

p.6, lines 137-138 Suggest modifying sentence as follows:

Starfish-like complexes of HA (Fig. 1A, black arrows) and isolated HA trimers (Fig. 1B, right panel) were present in Fluzone.

p.6, lines 140-142 Suggest adding "right panel" as follows:

Class averages of Flucelvax indicated the presence of HA-starfish-like complexes, isolated HA molecules, and NA heads which were identifiable by their tetrameric symmetry (Fig. 1D, right panel).

p.6-7, lines 145-147 The following sentence does not make it clear which sample is being referred to in the first half of the sentence:

Constituent HA molecules presented a prefusion peanut-shape, similar to the HA starfish structures presented in the class-averages of Flublok (Fig. 1F).

Suggest rewording as follows:

Constituent HA molecules in all four samples presented a prefusion peanut-shape, similar to the HA starfish structures presented in the class-averages of Flublok (Fig. 1F).

p.7, lines 169-170 The wording of the following phrase is unclear

(iii) HA-Fabs never clashed due to insufficient spacing that no axial rotation of HA could alleviate the steric clash

Suggest rewording as follows:

(iii) HA-Fabs do not clash due to sufficient space between the two HA-Fab complexes

p.8, line 186 Suggest replacing the word "microscope" with "microscopy"

p.14, lines 324-325 The meaning of the following sentence is not clear.
Although some HA starfish-like complexes did have eight HA trimers like the engineered HA ferritin nanoparticles 26,33, commercial vaccines contained full-length HAs.

Perhaps reword as follows (if this is the intended meaning):

Although some HA starfish-like complexes did have eight HA trimers like the engineered HA ferritin nanoparticles 26,33, commercial vaccines have the advantage of containing full-length HAs.

p.14, line 336 Suggest replacing "icosahedral symmetry vehicles" with "icosahedrally symmetric vehicles"

p.25, line 625, Fig. 2 legend, Suggest adding the word "are" as follows:

Within the central region of conditional steric clash, 57% of these configurations are free of steric clash.

Reviewer #2:

Remarks to the Author:

In their manuscript Myers and colleagues characterize the structural aspects of seasonal influenza virus vaccines. The work is interesting but there are several points that need the authors' attention.

Major points

- 1) The vaccine lots for all preparations used should be listed including years and strain composition. Also, not until the methods section we learn that the Fluzone vaccine was HD. Please specify that right away in the text.
- 2) The modeling regarding binding on HA was done with Fabs. But Antibodies are much bigger than Fabs. The modelling should be repeated taking full IgG into account.
- 3) It would be important to include quantification of pre-fusion versus post-fusion HA content for the vaccines.
- 4) It would be important to include quantification of NA content for the vaccines.

5) Figure 5: Why did C179 not bind?

6) Figure 5: None of the panels has a LOD.

Minor points

1) Throughout the manuscript: Whenever the virus is meant, 'influenza virus' should be used. 'Influenza' is the disease, 'influenza virus' is the causative agent. E.g. there is no 'influenza hemagglutinin', only 'influenza virus hemagglutinin'.

2) Line 51-53: Seasonal influenza puts the healthy population at risk as well. Please rephrase.

3) Line 53-54: This sentence should refer to seasonal influenza, not pandemic influenza.

4) Line 64 and throughout the manuscript. HA2 is not the stem, HA1 is not the head. Please look up the definitions and correct this throughout.

5) Line 75: Those are licensed vaccines, not 'vaccine designs'. Also, they are not synthesized. Virus is grown in cells and then inactivated. Please rephrase.

6) Line 85: 'flu' is a layman's term and should never be used in the scientific literature.

7) Lines 117/118: The lineages are called 'B/Victoria/2/87-like' and 'B/Yamagata/16/88-like'. Please

use proper nomenclature. There are plenty of Victoria viruses from both lineages.

8) Line 151: 'towards the camera'

9) Line 241: Define 'ELISA'.

10) Line 249: A coating control needs to be used to make sure there are really similar amounts on the plate. Coating efficiency can vary dramatically between formulations.

11) Line 267: Was the challenge virus matched to the vaccines?

12) Line 279-280, line 456, line 470: Why are words beginning with capitalized letters mid-sentence?

13) Line 306-307: Please define HIV and SARS-CoV-2. HIV-1 or HIV-2?

14) Line 375: Define 'PBS'.

15) Line 427: How much certainty can we have when only 58 discs are sampled?

16) Also define HRP, ABTS, MDCK-ATL, MEM, TPCK, DAB

17) Line 466: It is 'TPCK-treated trypsin'.

Reviewer #3:

Remarks to the Author:

In this paper the structural organization of HA molecules was analyzed by electron-microscopy based methods, comparing Fludac, Blublok, Flucelvax, and Fluzone vaccine compositions. Interestingly, each vaccine was found to induce a unique spectrum or even unique types of HA organization. Individual HA and starfish-like structures with up to 12 HA molecules attached via hydrophobic interactions of the tails of the HA stems were the most common. Interestingly, Fludac had spiked nanodisks displaying over 50 HA molecules along the complex perimeter. Notably, the nanodisks displayed HA at an angle allowing the best exposure of conserved stem epitopes. Improved display of stem epitopes was verified with mAbs and shown to be immunogenic in mice with improved anti-stem and heterosubtypic responses to the Fludac vaccine.

1. This study represents an important characterization of common commercial vaccine display of HA and the subdominant HA stem epitopes.
2. It would be good to know how whole virus compares to the various vaccines for both stem binding by the mAbs and for elicitation of stem or heterosubtypic antibodies (the assays in Fig. 5). For example, Andrews, et al., (Sci. Trans. Med.,2015) showed that anti-stem antibodies more readily bound soluble, recombinant HA than to whole virus. It would be good to know how this translates for the various vaccines in the context of this study.
3. Some discussion of the structural basis of immune-dominance of HA head over HA stalk epitopes seems warranted. That is, what does this study tell us about why HA head epitopes are so dominantly targeted by most vaccines (topic well covered I feel in this paper) or by infection with virus.

RESPONSE: We thank the reviewers for taking the time to provide helpful comments to improve our manuscript. Point-by-point responses to reviewer comments are detailed below.

REVIEWER COMMENTS

Reviewer #1 (Remarks to the Author):

The manuscript by Myers et al presents a novel examination of the various structural arrangements of Hemagglutinin (HA) within four commercial influenza vaccines. The authors demonstrate using cryo-electron tomography and Monte Carlo simulations with HA/stem-Fab complexes that the different HA structural presentations observed within the four vaccines lead to differences in predicted antibody binding availability of conserved HA stem epitopes. Logically one would expect that increased antibody binding availability of HA stem epitopes within vaccine formulations would lead to induction of higher levels of anti-stem antibodies, which might provide greater protection for future influenza variants. Surprisingly, the authors found in mice that induction of stem-targeted antibodies did not follow the same trend as stem epitope availability within the vaccine formulations. More importantly however, the vaccine formulation with the most exposed conserved stem epitopes, as presented within

a “spiked nanodiscs”, elicited the highest levels of heterologous HA subtype antibodies. This result, if confirmed by further studies with vaccines for other pathogens, would be helpful in generation of vaccines that induce broader immunity. This would be incredibly helpful in the design of a universal flu vaccine or a more broadly effective coronavirus vaccine.

The methodology presented is sound and the cryo-electron tomography structural analysis is state-of-the-art. There appears to be enough detail provided in the methods for the work to be reproduced. The major strength of the manuscript is that authors demonstrate (I believe for the first time) that the structural arrangement and presentations of epitopes within a vaccine formulation is a factor that can now be directly observed and correlated with vaccine effectiveness.

RESPONSE: As indicated following each point below, we have reviewed each of these items, we agree with them, and we have made each of the suggested revisions. Thank you to Reviewer #1 for your clear expression of your concerns and solutions.

Minor points and suggestions:

p.6, lines 137-138 Suggest modifying sentence as follows:

Starfish-like complexes of HA (Fig. 1A, black arrows) and isolated HA trimers (Fig. 1B, right panel) were present in Fluzone.

RESPONSE: We have implemented this suggestion in the revised manuscript in the last paragraph on page 6.

Starfish-like complexes of HA (Fig. 1a, black arrows) and isolated HA trimers (Fig. 1b, right panel) were present in Fluzone HD.

p.6, lines 140-142 Suggest adding “right panel” as follows:

Class averages of Flucelvax indicated the presence of HA-starfish-like complexes, isolated HA molecules, and NA heads which were identifiable by their tetrameric symmetry (Fig. 1D, right panel).

RESPONSE: We have implemented this suggestion in the revised manuscript in the last paragraph on pages 6-7.

Class averages of Flucelvax indicated the presence of HA-starfish-like complexes, isolated HA molecules, and NA heads which were identifiable by their tetrameric symmetry (Fig. 1d, right panel).

p.6-7, lines 145-147 The following sentence does not make it clear which sample is being referred to in the first half of the sentence:

Constituent HA molecules presented a prefusion peanut-shape, similar to the HA starfish structures presented in the class-averages of Flublok (Fig. 1F).

Suggest rewording as follows:

Constituent HA molecules in all four samples presented a prefusion peanut-shape, similar to the HA starfish structures presented in the class-averages of Flublok (Fig. 1F).

RESPONSE: This region of the text was revised upon adding new text regarding prefusion/postfusion HA (new Supplementary Fig. 4) This sentence was removed and the prefusion component was addressed in new text on page 7.

To address if individual HA molecules were in prefusion or post-fusion conformation, individual HA trimers were manually picked to select proteins that were sufficiently isolated from other entities in the EM images. These individual HA trimers were averaged to improve resolution, and then measured for comparison to previously reported structures. We found that all influenza vaccines yielded HA profiles that were consistent with prefusion HA conformation, based upon the height of the HA trimer, and based upon the shape profile (Supplementary Fig. 4).

p.7, lines 169-170 The wording of the following phrase is unclear

(iii) HA-Fabs never clashed due to insufficient spacing that no axial rotation of HA could alleviate the steric clash

Suggest rewording as follows:

(iii) HA-Fabs do not clash due to sufficient space between the two HA-Fab complexes

RESPONSE: We have implemented this suggestion in the revised manuscript on page 8.

(iii) HA-Fabs do not clash due to sufficient space between the two HA-Fab complexes (Fig. 2b).

p.8, line 186 Suggest replacing the word “microscope” with “microscopy”

RESPONSE: We have implemented this suggestion.

p.14, lines 324-325 The meaning of the following sentence is not clear.

Although some HA starfish-like complexes did have eight HA trimers like the engineered HA ferritin nanoparticles 26,33, commercial vaccines contained full-length HAs.

Perhaps reword as follows (if this is the intended meaning):

Although some HA starfish-like complexes did have eight HA trimers like the engineered HA ferritin nanoparticles 26,33, commercial vaccines have the advantage of containing full-length HAs.

RESPONSE: Revision implemented, and then we undertook additional edits here for further clarification of the main point: (page 15, second paragraph of the revised manuscript).

Different from HA starfish-like complexes found in the engineered HA ferritin nanoparticles^{28,36}, commercial vaccines present a medley of arrangements for a given number of HAs, which are made possible by full-length HAs being embedded in a lipid component.

p.14, line 336 Suggest replacing “icosahedral symmetry vehicles” with “icosahedrally symmetric vehicles”

RESPONSE: We have implemented this suggestion in the revised manuscript in the first paragraph of page 16.

These larger spiked nanodiscs also contain more HA trimers than most nanoparticle scaffolds, perhaps equivalent to icosahedrally symmetric vehicles.

p.25, line 625, Fig. 2 legend, Suggest adding the word “are” as follows:
Within the central region of conditional steric clash, 57% of these configurations are free of steric clash.

RESPONSE: We have implemented this suggestion in the revised manuscript for Fig. 2 legend (page 30).

Within the central region of conditional steric clash, 57% of these configurations are free of steric clash.

Reviewer #2 (Remarks to the Author):

In their manuscript Myers and colleagues characterize the structural aspects of seasonal influenza virus vaccines. The work is interesting but there are several points that need the authors’ attention.

Major points

1) The vaccine lots for all preparations used should be listed including years and strain composition. Also, not until the methods section we learn that the Fluzone vaccine was HD. Please specify that right away in the text.

RESPONSE: The first mention of Fluzone has been updated to be describe it as “Fluzone High Dose (HD)”, and we have updated all references to “Fluzone” in the revised manuscript to “Fluzone HD”. Lot number information for all influenza vaccine samples has been added to our table of vaccine information: Supplementary Fig. 1 in the revised manuscript.

2) The modeling regarding binding on HA was done with Fabs. But Antibodies are much bigger than Fabs. The modelling should be repeated taking full IgG into account.

RESPONSE: We agree the Fc domain of full IgG could impact the sterics of binding. The flexibilities of linkers to the individual Fab domains make it difficult to evaluate the impact of the full-length IgG, but we have added text and supplementary images to present the issue to the reader, and some coarse-grained conclusions. We added citations describing the ensemble of conformations that full length antibody samples, but we are unfortunately not able to reproduce the rigor of that work within the context of this manuscript. However, we did add supplemental analysis to visualize full-length IgG binding steric constraints to HA stem epitopes (Supplementary Fig. 11).

(New) Supplementary Fig. 11:

Supplementary Figure 11. Visualization of full-length IgG binding steric constraints. HA trimers (green and blue) modeled in close proximity but with enough space to permit Fab domain binding are illustrated for HA trimers that are parallel (top), or perpendicular (bottom). Three different PDB structures of full-length IgG are illustrated in each of the three columns (ribbon representation). Bound Fab for a second IgG is illustrated in gray space filling representation. The bases of the parallel HA molecules are 185 Å apart, and the bases of perpendicular HA molecules are 80 Å apart. Considering the flexibility available to the second IgG, the conformations given by PDB IDs 1IGT and 1HZH on perpendicular HA trimers are capable of avoiding steric clashes with a second IgG, while the conformation given by 1IGY is likely to be in steric clash. For parallel HA trimers, all three IgG conformations illustrated are likely to be in steric clash with a second IgG binding event.

RESULTS: (page 7, last paragraph)

We modeled the interaction with fragment antigen binding (Fab) domain instead of full IgG, thus our findings represent an upper bound to the number of sterically permitted binding conformations. IgG molecules have flexibility between domains³¹, which allows them to deviate from a rigid “Y” shaped conformation. This flexibility enables the approximation of using a Fab to represent the IgG in many bound conformations, but not all, which is further addressed in *Discussion*.

DISCUSSION:

PAGE 16, second paragraph

IgG is comprised of three rigid domains: two Fab domains which each separately bind antigen, and an Fc domain that binds receptors and immune effectors. Bivalent binding by IgG can occur when epitopes are spaced between 4-16 nm apart³², resulting in high avidity binding. When antigens are densely spaced but not compatible with bivalent binding, Fc and Fab domains may sterically hinder IgG from binding neighboring antigens. The linkers between the three domains in IgG have preferred orientations, but are flexible to a degree that varies by antibody subtype³¹.

Since the bound Fab domain is the only fragment of the IgG rigidly positioned by antigen binding, we employed a Monte Carlo method ...

PAGE 17, first paragraph

Our simulations address only the rigid Fab domain, but the impact of full IgG or even IgM should also be considered. In supplementary work (Supplementary Fig. 11) we first used the 5 crystal structures of full-length IgG (PDB IDs 1HZH, 1IGT, 1IGY, 5DK3, 6GFE), which we aligned to representative poses of HA trimers illustrated in Fig. 2. When HA trimers were positioned at a 90 degree angle (Fig. 2b), we were surprised to find that the off-center attachment site of the Fc-Fab linker as it connects to the Fab domain often avoided steric clashes that were anticipated due to close proximity of bound Fab domains. As HA trimer orientations became parallel and the orientation of bound Fab domains approached 180 degrees, clashes were observed with either the Fc domain or the unbound Fab domain (Supplementary Fig. 11). These instances, where Fab domains were just beyond the range of bivalent binding yet still oriented in opposite directions, were the conformations most affected by the presence of the full-length IgG. Still, quantitation of this effect would be specific to antibody type and subtype, and would require rigorous knowledge of the linker length and flexibility, such as described for IgG1³¹ and IgD⁴⁰. In an intriguing case where IgG structure was resolved in bivalently-bound complexes of 4-, 5-, and 6-membered rings with its CD20 antigen, Fc domains were poorly resolved due to substantial flexibility even when both Fab domains were immobilized by antigen⁴¹. We conclude that while the physical presence of Fc and unbound Fab domains must play a role in the sterics of binding dense antigen, the significance is minimal for most binding orientations of this HA model system due to the Fc-Fab linker positioning and flexibility.

3) It would be important to include quantification of pre-fusion versus post-fusion HA content for the vaccines.

RESPONSE: While we agree with the issue that is raised by quantifying pre-fusion vs. post-fusion for influenza, the impact of contaminating post-fusion structures in influenza vaccine has not proved to be as consequential as for other viruses that have surface glycoproteins that exist as pre and post-fusion states on the virus surface. These include the Fusion (F) glycoprotein of respiratory syncytial virus vaccine (RSV) or the Spike (S) protein of SARS-CoV-2. We lacked reagents or methods to pursue this in great detail because in the field there is a lack of antibodies that are robustly proven to be specific for the post-fusion state of trimeric HA.

However, a number of studies (e.g. Harris et al., 2006 PNAS) have indicated that viral full-length HA is generally in the prefusion state. In addition, stem antibodies such as FI6v3, CR6261 and C179 are specific for the stem of HA in the prefusion state (e.g. Yassine et al., 2015 Nat Med).

Thus, our new revised sandwich ELISA analysis with stem antibodies indicated that the prefusion state is present in the vaccines (Fig. 5a) as does electron microscopy that indicated a prefusion peanut shape for HA molecules (Fig. 1).

However, to add additional analysis concerning the prefusion state of HA we leveraged how negative stain electron microscopy can visualize each protein. First, we added full-field micrographs of each influenza vaccine studied as new Supplementary Fig. 3 (see below)

Supplementary Figure 3. Negative stain EM images of influenza vaccines. Representative electron micrographs of the four influenza vaccines, illustrating the differential organization of HA proteins for (a) Fluzone HD, (b) Flucelvax, (c) Flublok, and (d) Fluad. Scale bar is 50 nm.

Second, for each vaccine we performed additional image analysis of individual HA trimers in Supplementary Fig. 4 (see below). Third, we added these additional results to the RESULTS section.

The additional 2D image analysis indicated HA in a prefusion state for the four vaccines (Fluzone HD, Flucelvax, Flublok, Fluad) (new Supplementary Fig. 4).

Supplementary Figure 4. Class averages of individual HA trimers in vaccines. (a) Ribbon-diagram depiction for prefusion (PDB ID 1HA0) postfusion (PDB ID 1HTM) HA structures. To compare x-ray crystallography-derived structures to EM images, PDB coordinates were used to calculate a volume with Chimera, then 2D projections were made with EMAN2 for (b) prefusion and (c) postfusion HA. (d) Reference structure of full-length HA in a lipid micelle, determined by single particle EM to 4.2 Å (EMD-0237) is given for comparison. (e-h) 2D-Class averages from negative-staining electron microscopy manually picked, isolated HAs in vaccines (e) Fluzone, (f) Flucelvax, (g) Flublok, and (h) Fluad. (i) The HA density profile is plotted, where the positive x-axis traces along the threefold symmetry axis of HA trimers, starting from the base, continuing through the stem, then head, to the top of the HA trimer. (j) Total length of HA trimers, in the direction from the stem to the head. Scale bars are 50 Å.

Additional text was added to the revised manuscript Results section as follows: (page 7, first paragraph)

To address if individual HA molecules were in prefusion or post-fusion conformation, individual HA trimers were manually picked to select proteins that were sufficiently isolated from other entities in the EM images. These individual HA trimers were averaged to improve resolution, and then measured for comparison to previously reported structures. We found that all influenza vaccines yielded HA profiles that were consistent with prefusion HA conformation, based upon the height of the HA trimer, and based upon the shape profile (Supplementary Fig. 4).

4) It would be important to include quantification of NA content for the vaccines.

RESPONSE: To quantitate NA, we obtained recombinant protein of N1 NA and performed western blot analysis of the vaccines and compared them to binding of known dilutions of recombinant NA protein. We focused on N1 NA because H1N1 viruses caused pandemics in 1918 and 2009. We found that Fludac had the highest NA content among commercial vaccines, although the significance of this observation is difficult to assess because the total amount of NA in Fludac was 0.16ug of NA for every 1ug HA. These additional NA results are implemented as text additions to the Results, Discussion, and Methods and as a new Supplementary Fig. 5 (see below).

Supplementary Figure 5. Quantification of N1 NA in commercial vaccines by western blot. In panel *a-b*, decreasing concentrations of recombinant NA protein was applied (120, 60, 30, 15, and 7.5 ng) in lanes 2-6. Remaining lanes were loaded with influenza vaccines and rHA at 0.3 ug H1 HA per sample. Blots were imaged for (a) 1 sec or (b) 30 sec. Uncropped blots in Source Data. A standard curve (black line) was generated for NA concentration (c) based on densitometry values from panel *a* (black dots) and shown with a 90% confidence interval (black dotted lines), the goodness of fit statistic (R^2) for the standard curve is also displayed. The concentration of NA in the Fludac vaccine was extrapolated from the curve (blue dot). A control blot (d) was probed with a polyclonal antibody to H1 HA and exposed for 1sec to ensure there was adequate protein transfer for detection. Focus was on N1 NA because H1N1 viruses caused pandemics in 1918 and 2009.

RESULTS: This additional text was added (page 11, second paragraph).

NA quantitation. NA presence in influenza vaccines has increasingly been the focus of influenza vaccine design strategies. Because we observed individual NA tetramers in negative stain EM images, we sought to quantify bulk N1 NA content as proof-of-concept across these four influenza vaccine formulations. Focus was on N1 NA because H1N1 viruses are both seasonal and have caused pandemics in 1918 and 2009. Using western blot, we compared NA content in the vaccines

to known dilutions of recombinant NA. While NA was observed by EM in all influenza vaccine formulations except Flublok, Fluad contained the highest concentration of NA (Supplementary Fig. 5). From the absolute concentration of NA, we found that the relative concentration of NA in Fluad was approximately 0.16 ug of NA for 1 ug HA, or equivalent to 2.4 ug NA per influenza strain, per vaccine dose. Meanwhile, all other vaccines contain less than our detection threshold of 0.025 ug NA per 1 ug HA (Supplementary Fig. 5).

5) Figure 5: Why did C179 not bind?

RESPONSE: When C179 is utilized as a capture antibody it was able to bind Flublok, Fluad, and Fluzone HD but not Flucelvax. The monoclonal stem antibodies we utilize vary in their specific binding footprint so difference in binding between the three are not unexpected. In particular, C179 was originally selected for binding to H2 HA and found to be cross-reactive with H1 HA (Okuno et al., "A common neutralizing epitope conserved between the hemagglutinins of influenza A virus H1 and H2 strains." J Virol. 1993.) therefore it's possible that its binding footprint has less specificity than the other two monoclonals tested and could be more easily perturbed. However, the original direct ELISA was replaced in the revised manuscript by a sandwich ELISA and this improved the binding of C179 to the vaccines See revised Fig. 5a below.

Fig. 5. Commercial vaccine stem-epitope availability and cross-reactive immunogenicity. (a) ELISA binding analysis of commercial vaccines (Fluad, Flublok, Fluzone HD, Flucelvax) by HA stem monoclonal antibodies (FI6v3, CR6261, C179).

METHODS: (Additional Method added).

ELISA for Epitope Accessibility. Stem monoclonal antibodies (FI6v3, CR6261, and C179 provided by the Vaccine Research Center, NIH, Bethesda, MD) were applied to microtiter plates (immulon 2 B, Thermo Fisher) at 1.25 µg/ml to act as a capture antigen and incubated overnight at 4°C. Plates were washed between all experimental step with PBS plus 0.1% tween 20 (405 TS ELISA plate washer, Agilent Technologies). Blocking (1% omniblok, AmericanBio, Inc., and 0.1% tween 20 in PBS) occurred for 30mins at room temperature prior to application of commercial vaccines or rHA at 1.25 µg/ml of H1 HA or saline for 2hrs before washing and probing with a rabbit polyclonal antibody to H1 HA (PS6000, Protein Sciences) for an additional 2hrs. Plates were than incubated at 37°C for 1hr with a goat anti-rabbit horseradish peroxidase (HRP) conjugated secondary antibody (goat anti-rabbit IgG (H+L), Thermo Fisher). Colorimetric detection occurred for 15 min at room temperature and absorbance was read at 405 nm (1-Step 2,2'azinobis-[3-ethylbenzothiazoline-6-sulfonic acid] diammonium salt (ABTS), Thermo Fisher) at 405 nm. Samples were run in triplicate. A background control was subtracted and the data is displayed as mean with standard error of the mean (SEM) bars using Prism (GraphPad).

6) Figure 5: None of the panels has a LOD.

RESPONSE: Revised Fig. 5a displays ELISA data that has been PBS background subtracted and now includes a saline group to show baseline binding.

Figures 5 c-f are showing ELISA data that has been analyzed with a statistical method to determine when our sera are no longer different from the saline control group (end point titer). This method doesn't require PBS background subtraction. When calculated, no difference from the saline control sera at the highest concentration tested would be displayed as a value of ~ 2 on the graphs shown. We have updated the methods section to include this information. The dilution curves for these figures, with the saline controls included, can be found in the Supplementary Figs. 7, 8, 9.

In the Methods section under the ELISA for Epitope Accessibility section we provide additional details:

Endpoint titer levels were statistically defined per plate⁵⁶ as the lowest sera dilution still differing from the saline group, an endpoint titer of 2.01 Log(reciprocal dilution factor) indicates no difference from the saline group at the highest sera concentration tested.

Minor points

1) Throughout the manuscript: Whenever the virus is meant, 'influenza virus' should be used. 'Influenza' is the disease, 'influenza virus' is the causative agent. E.g. there is no 'influenza hemagglutinin', only 'influenza virus hemagglutinin'.

RESPONSE: We have implemented this in the revised manuscript.

2) Line 51-53: Seasonal influenza puts the healthy population at risk as well. Please rephrase. 3) Line 53-54: This sentence should refer to seasonal influenza, not pandemic influenza.

RESPONSE: We have added a sentence and a reference to address this point within the first paragraph of the introduction, allowing the first of three sentences to address at-risk individuals for seasonal influenza, the second for "prime-aged" individuals assumed to generally have healthy immune systems afflicted by seasonal influenza, and the third for pandemic influenza afflicting everyone.

INTRODUCTION (first paragraph):

At-risk individuals that may be immunologically suppressed due to age or specific health issues can develop life threatening infections². Yet even prime aged individuals may fall victim to seasonal influenza, especially with the lack of supportive care for these infections³. When new zoonotic strains are created by reassortment between animal and human lineages, antigenically novel influenza viruses have caused pandemics which have put even the healthy populations at great risk⁴

4) Line 64 and throughout the manuscript. HA2 is not the stem, HA1 is not the head. Please look up the definitions and correct this throughout.

RESPONSE: We have reviewed our application of the terms and revised the text to use "head" when we refer to the structure which is top of the HA molecule, which happens to be most physically exposed on the surface of virus. Then the "stem" region is used to describe the structure at the opposite end of HA from the head, and nearer to the membrane. We have revised references to HA1 and HA2 to refer to their peptide sequence within the structure,

rather than locations in the structure. The introduction was revised, and the entire manuscript was reviewed for terms “HA1”, “HA2”, “head”, and “stem”. Most of the time, it was appropriate to remove references to HA1 and HA2, and then use head and stem.

INTRODUCTION (second paragraph on page 3):

HA1 is the N-terminal fragment and comprises the “head” domain, which is the membrane-distal region of the 13 nm tall viral glycoprotein. HA1’s protein sequence also contributes to the “stem” domain at the opposite end of the HA protein from the head and nearer to the viral envelope. HA2 is the C-terminal fragment of HA0 cleavage, and all residues of HA2 contribute to the stem domain. HA2 contributes the most conserved amino acids in HA¹², which perform the act of fusing host and viral membranes during viral entry. Because the HA head is both immunodominant and highly variable, yearly reformulation of influenza vaccines is required to ensure that HA antigenic variation does not escape vaccine-elicited antibodies^{13,14}.

5) Line 75: Those are licensed vaccines, not ‘vaccine designs’. Also, they are not synthesized. Virus is grown in cells and then inactivated. Please rephrase.

RESPONSE: We have rephrased the text to improve accuracy in the revised manuscript (Introduction, first paragraph on page 4):

To avoid the problem of egg-adapted HA mutations, new strategies have been developed to manufacture vaccines directly from mammalian cells infected with influenza²⁰

6) Line 85: ‘flu’ is a layman’s term and should never be used in the scientific literature.

RESPONSE: We have now corrected this. There is now no use of “flu” in this manuscript.

7) Lines 117/118: The lineages are called ‘B/Victoria/2/87-like’ and ‘B/Yamagata/16/88-like’. Please use proper nomenclature. There are plenty of Victoria viruses from both lineages.

RESPONSE: We agree with this point of being exact in communication, but in this particular case the manufacturers have freedom to elect between related strains that best fit their process. We revised our main text to remove the reference to the specific strains, but then direct readers to Supplementary Fig. 1 that lists the exact composition of the vaccines.

RESULTS (first paragraph of Results section on page 5).

Of the four commercial vaccines studied, Fluvad and Fluzone High Dose (HD) are trivalent, containing HA antigens for H1 and H3 subtypes for influenza A virus, and B/Victoria lineage for influenza B virus. Flublok and Flucelvax are quadrivalent vaccines due to their composition including an additional influenza B virus B/Yamagata strain. Specific influenza strain composition for each vaccine is given in Supplementary Fig. 1a.

8) Line 151: ‘towards the camera’

RESPONSE: We revised this text as advised, but then further revised the surrounding sentence aiming for additional clarity through brevity: (page 7, second paragraph).

Many HA proteins on these complexes were visible as top-views that visualized the HA three-fold symmetry axis as a triangular arrangement of HA protomers.

9) Line 241: Define ‘ELISA’.

RESPONSE: We have fixed this. (page 11. Last paragraph)

To probe these differences, we assessed binding (*in vitro* display) by enzyme linked immunosorbent assay (ELISA) using stem antibodies.

10) Line 249: A coating control needs to be used to make sure there are really similar amounts on the plate. Coating efficiency can vary dramatically between formulations.

RESPONSE: We thank the reviewer for pointing out this concern, we conducted a follow up experiment using a sandwich ELISA format, which coats the wells with the mAb as a capture antibody before vaccines are plated. By alternating the protocol, it minimizes any difference in coating efficiency that the vaccine formulations have since all wells are identically coated initially. The updated experimental results are reported in the paper. The revised Fig. 5a is below.

Fig. 5. Commercial vaccine stem-epitope availability and cross-reactive immunogenicity. (a) ELISA binding analysis of commercial vaccines (Fluad, Flublok, Fluzone HD, Flucelvax) by HA stem monoclonal antibodies (FI6v3, CR6261, C179).

RESULTS: (last paragraph starting page 11, pages 11-12)

Anti-stem monoclonal antibodies FI6v3, CR6261, and C179 were used as capture antibodies to detect differences in the availability of stem epitopes in the commercial influenza vaccines Fluad, Flublok, Fluzone HD, and Flucelvax (Fig. 5a). All vaccines contained HA protein capable of binding epitopes to anti-HA stem antibodies FI6v3 and CR6261. However, while all vaccines showed similar levels of binding to FI6v3, after normalizing the formulations for HA concentration, there were differences in binding to CR6261 indicating that both Flublok and Fluad had more stem epitope binding availability than Fluzone HD and Flucelvax. These differences were more pronounced when capture antibody C179 was utilized, C179 bound significantly more Flublok than any other vaccine (Fluad, Fluzone HD, and Flucelvax). C179 was particularly poor at binding the H1 HA in Flucelvax, showing no difference from saline (Fig. 5a).

11) Line 267: Was the challenge virus matched to the vaccines?

RESPONSE: We have added text to clearly state that the challenge virus was matched to the vaccines (page 13 first paragraph).

...we challenged mice with a A/Michigan/45/2015 (H1N1) virus that was a match to the H1 vaccine component and looked for differences in survival, weight loss (Fig. 6)...

12) Line 279-280, line 456, line 470: Why are words beginning with capitalized letters mid-sentence?

RESPONSE: Capitalization issues have been resolved.

13) Line 306-307: Please define HIV and SARS-CoV-2. HIV-1 or HIV-2?

We have added these definitions (Discussion, page 14)

...multitude of different viral proteins such as influenza HA, human immunodeficiency virus 1 (HIV-1) Env, and severe acute respiratory syndrome coronavirus 2 (SARS-CoV-2) spike^{28,36-38}

14) Line 375: Define 'PBS'.

RESPONSE: We have fixed this in the revised Methods section.

Samples (e.g. Flucelvax, Fluzone HD, Flublok, and Fluad) were diluted in phosphate-buffered saline (PBS)...

15) Line 427: How much certainty can we have when only 58 discs are sampled?

RESPONSE: To empirically measure certainty of this dataset, we undertook direct, manual measurements of the bilayer thickness (the result derived from these 58 nanodiscs). Using the subset of these 58 where the particle side was approximately aligned with the viewing angle, (29 nanodiscs), we measured the bilayer distance directly for each particle, then we reported the average and standard deviation:

RESULTS: (page 10)

Because the resolution of the subtomogram average was anisotropic, we separately estimated the error of the bilayer width by manually measuring the cross-sectional distance between bilayer peaks in particles where the bilayer was viewed from the side, and we found the average and standard deviation of this distance to be $50.6 \pm 4.5 \text{ \AA}$.

16) Also define HRP, ABTS, MDCK-ATL, MEM, TPCK, DAB

RESPONSE: We have fixed these in revised Methods. We found we also needed to add: "ribonucleoprotein (RNP)", "fragment, antigen binding (Fab)"

17) Line 466: It is 'TPCK-treated trypsin'.

RESPONSE: We have fixed this in revised Methods.

...with 1 $\mu\text{l/ml}$ N-p-tosyl-L-phenylalanine chloromethyl ketone (TPCK)-treated trypsin

Reviewer #3 (Remarks to the Author):

In this paper the structural organization of HA molecules was analyzed by electron-microscopy based methods, comparing Fluad, Blublok, Flucelvax, and Fluzone vaccine compositions. Interestingly, each vaccine was found to induce a unique spectrum or even unique types of HA organization. Individual HA and starfish-like structures with up to 12 HA molecules attached via hydrophobic interactions of the tails of the HA stems were the most common. Interestingly, Fluad had spiked nanodisks displaying over 50 HA molecules along the complex perimeter. Notably, the nanodisks displayed HA at an angle allowing the best exposure of conserved stem epitopes. Improved display of stem epitopes was verified with mAbs and shown to be immunogenic in mice with improved anti-stem and heterosubtypic responses to the Fluad vaccine.

1. This study represents an important characterization of common commercial vaccine display of HA and the subdominant HA stem epitopes.

2. It would be good to know how whole virus compares to the various vaccines for both stem binding by the mAbs and for elicitation of stem or heterosubtypic antibodies (the assays in Fig. 5). For example, Andrews, et al., (Sci. Trans. Med., 2015) showed that anti-stem antibodies more readily bound soluble, recombinant HA than to whole virus. It would be good to know how this translates for the various vaccines in the context of this study.

RESPONSE: To address this point we conducted an additional set of experiments to compare the binding of stem mAbs and of sera from vaccinated animals to recombinant HA and to HA displayed on virus. Our results were consistent with Andrews (2015), in that we found that all antibodies tested (mAb and vaccine sera) bound higher to the recombinant HA than to the HA on virus. This additional analysis is noted in the text and data is presented in Supplementary Fig. 6 (see below).

Supplementary Figure 6. Binding to HA displayed on virus. (a) Monoclonal stem antibodies (FI6v3 square, CR6261 circle, and C179 triangle) bind to increasing concentrations of virus. (b) The difference in binding on virus and rHA is shown for monoclonal stem antibodies from panel a. (c) Rabbit immunization with injections occurred on day 0, 14, and 42 and bleeds occurred on day 0, 14, 42, and 56. (d) The difference between binding to virus and rHA by immunized rabbit sera from day 56.

RESULTS: (page 11, last paragraph)

Recombinant H1 protein was used as a positive binding control since both stem antibodies and vaccinated animal sera bind to it more readily than HA on virus (Supplementary Fig. 6).

3. Some discussion of the structural basis of immune-dominance of HA head over HA stalk epitopes seems warranted. That is, what does this study tell us about why HA head epitopes are so dominantly targeted by most vaccines (topic well covered I feel in this paper) or by infection with virus.

RESPONSE: Indeed, such discussion is well warranted, though much room is left for future work. We have added the following text (see below) to the discussion to provide perspective to readers about immune-dominance of the variable HA head, and how we have searched for answers within the space of existing vaccines rather than the larger search space of novel designed antigens. Additional text was added in the Discussion section (page 18, second paragraph) as follows:

Dominance of the immune response to highly variable epitopes on the HA head is a known challenge in influenza vaccine design, and is an important factor in limiting the effectiveness of immunity due to natural infection⁴³. Whether or not HA stem epitopes are selected against in B cell maturation due to the nature of the HA protein itself is an ongoing question. While not outright solving this problem, we demonstrate in this work that even among full-length HA antigens that include the variable head epitopes, improvements can be made in targeting a greater fraction of the immune response to conserved stem epitopes (Fig. 5). Even though designed antigens may illicit only a stem epitope response, antibodies targeting the head typically neutralize virus much more effectively when the viral strain is matched to the vaccine⁴³. Thus it is desirable to understand how a single vaccine may elicit both head and stem antibodies in a way that neither response dominates over the other. Here we have both described how four influenza vaccines differ in their organization structure of antigen, as well as the nature of their response, hopefully pointing the way for future vaccines to build upon the best of these characteristics.

Reviewers' Comments:

Reviewer #2:

Remarks to the Author:

The authors addressed the reviewers' comments well. I appreciate that additional experiments were done.

Reviewer #3:

Remarks to the Author:

All concerns have been adequately addressed.